# PIFiA: self-supervised approach for protein functional annotation from single-cell imaging data

Anastasia Razdaibiedina[1,2,3], Alexander Brechalov[1,2], Helena Friesen[2], Mojca Mattiazzi Usaj[2,6], Myra Paz David Masinas[2], Harsha Garadi Suresh[2], Kyle Wang[1,2], Charles Boone [1,2,4✉], Jimmy Ba [3,5✉] & Brenda Andrews [1,2✉]

## Abstract

Fluorescence microscopy data describe protein localization patterns at single-cell resolution and have the potential to reveal whole-proteome functional information with remarkable precision. Yet, extracting biologically meaningful representations from cell micrographs remains a major challenge. Existing approaches often fail to learn robust and noise-invariant features or rely on supervised labels for accurate annotations. We developed PIFiA (Protein Image-based Functional Annotation), a self-supervised approach for protein functional annotation from single-cell imaging data. We imaged the global yeast ORF-GFP collection and applied PIFiA to generate protein feature profiles from single-cell images of fluorescently tagged proteins. We show that PIFiA outperforms existing approaches for molecular representation learning and describe a range of downstream analysis tasks to explore the information content of the feature profiles. Specifically, we cluster extracted features into a hierarchy of functional organization, study cell population heterogeneity, and develop techniques to distinguish multi-localizing proteins and identify functional modules. Finally, we confirm new PIFiA predictions using a colocalization assay, suggesting previously unappreciated biological roles for several proteins. Paired with a fully interactive website (https://thecellvision.org/pifia/), PIFiA is a resource for the quantitative analysis of protein organization within the cell.

**Keywords** Self-supervised; Machine Learning; Single-cell; Imaging; Protein
**Subject Categories** Methods & Resources; Organelles

## Introduction

Recent progress in high-throughput microscopy and computational image analysis has catalyzed large-scale efforts to quantitatively describe single-cell biology (Cho et al, 2022; Chong et al, 2015; Mattiazzi Usaj et al, 2020; Mattiazzi Usaj et al, 2016; Thul et al, 2017; Thul and Lindskog, 2018). Advances in quantitative analysis of large-scale image datasets have been driven by the development of algorithms for protein localization prediction, which have been used for automated drug screening, and extracting morphological profiles from cell images (Kraus et al, 2017; McQuin et al, 2018). Computational methods enable efficient analysis of millions of single-cell images by extracting morphological information in an unbiased quantitative form. However, generating meaningful numerical features from single-cell images remains a significant challenge. Cells in micrographs typically exhibit a variety of shapes and positions, while noise levels and pixel intensities can also vary between images, making it difficult to develop algorithms that extract functionally rich patterns while ignoring irrelevant information (Kraus et al, 2017). For instance, early machine-learning approaches relied on hand-engineered feature sets extracted from images, such as cell texture and shape, which were often difficult to select and not transferable to other datasets or tasks (Chong et al, 2015). Ideally, a computational workflow would map single cells and proteins to robust numerical representations, enabling analysis of the spatial organization of the cell in an objective way.

More recently, single-cell images have been analyzed using deep learning methods, which overcome the limitations associated with hand-engineered feature sets by learning the optimal feature representations directly from pixel level data (Chong et al, 2015; Grys et al, 2017). The first machine-learning approaches for automated analysis of proteins' localization patterns were supervised (Kraus et al, 2017). Such methods were trained on a specific classification task, such as predicting cellular compartments from the input images (Grys et al, 2017; Kraus et al, 2017). While supervised methods achieve state-of-the-art performance in their target tasks, they require manual annotation of images used for training, which is time-consuming and expensive. In one of the efforts to accelerate label collection, the Human Atlas Project leveraged crowd-sourcing on a large scale, involving thousands of video games players for microscopy image annotation (Sullivan et al, 2018). However, manual label assignment is still not practical for imaging datasets containing millions of single-cell micrographs (Huh et al, 2003). In addition, human-labeled standards may reflect

[1]Department of Molecular Genetics, University of Toronto, Toronto, ON, Canada. [2]The Donnelly Centre, University of Toronto, Toronto, ON, Canada. [3]Vector Institute for Artificial Intelligence, Toronto, ON, Canada. [4]RIKEN Center for Sustainable Resource Science, 2-1 Hirosawa, Wako, Saitama, Japan. [5]Department of Computer Science, University of Toronto, Toronto, ON, Canada. [6]Present address: Department of Chemistry and Biology, Toronto Metropolitan University, Toronto, ON, Canada. ✉E-mail: charlie.boone@utoronto.ca; jba@cs.toronto.edu; brenda.andrews@utoronto.ca

the biases of an individual annotator and can preclude identification of subtle or incompletely penetrant phenotypes (Kraus et al, 2016; Lu et al, 2019). These problems motivated the development of methods that do not rely on supervised annotations during training (Lu et al, 2019; Moshkov et al, 2024).

An emerging alternative to supervised methods for biological image analysis involves self-supervised approaches, which do not require manually assigned categories during training (Chen et al, 2020; Jenni and Favaro, 2018; Jing and Tian, 2020). Instead, self-supervised learning models define a training objective, or pretext task, using structural information from the data itself (Jaiswal et al, 2020; Jing and Tian, 2020; Kolesnikov et al, 2019). In the context of self-supervised training, features learned with the pretext task should encapsulate information from the images that is useful for downstream applications, such as the discovery of common localization patterns by clustering analysis (Sullivan et al, 2018). Recently, self-supervised and weakly-supervised methods based on auto-encoders have been used for representation learning on cellular data (Guo et al, 2020; Kobayashi et al, 2022; Zaritsky et al, 2021). For example, weakly supervised learning with convolutional neural networks has been successfully applied for modeling associations between images and treatments, significantly improving performance over classical features (Moshkov et al, 2024). Autoencoder-based models are trained by compressing an image into the latent space (encoding), and subsequent image reconstruction (decoding) (Kingma and Welling, 2013). The encoding of the image in the latent space is then used as its representation. For instance, Paired Cell Inpainting (Lu et al, 2019), a self-supervised approach developed for analysis of yeast fluorescent micrographs, encodes several imaging channels to predict the appearance of a fluorescently-tagged protein in a target cell. Another autoencoder-based method developed for human cell data, *cytoself* (Kobayashi et al, 2022), trains a vector-quantized variational autoencoder (Kingma and Welling, 2013; Van Den Oord et al, 2017) (VQ-VAE) to reconstruct fluorescent signals of tagged proteins. Self-supervised learning with autoencoder-based approaches has also been applied for the analysis of human microglia data (Guo et al, 2020) and extraction of feature profiles predictive of cell metastatic potential (Zaritsky et al, 2021). One of the main disadvantages of auto-encoders is their difficulty in implementation and training challenges, as well as imperfect decoding (Chen et al, 2020; Jenni and Favaro, 2018; Kolesnikov et al, 2019). While *cytoself* and Paired Cell Inpainting achieved strong performance with decoder-based representations, replicating these networks on other datasets may be prohibitively complex. In this study, we asked whether other characteristics of microscopy data could be leveraged as self-supervised objectives to learn high-quality image representations with a relatively simple convolutional neural network.

Another challenge related to learning image-based features lies in their downstream analysis and interpretation. Current approaches typically extract representations with various machine learning methods and perform downstream analysis using clustering and tSNE/UMAP projections. (Kobayashi et al, 2022; Kraus et al, 2017; Sullivan et al, 2018). However, there are no clear rules for more nuanced biological analysis, including analysis of extracted features for different levels of cellular organization, or high-confidence identification of protein functional modules. In general, data-backed guidelines on hyperparameter selection, which enable biologically meaningful clustering and consider the scale of

cellular organization, are needed. Also, current molecular representation learning approaches generally lack methodologies that can characterize protein function by quantifying cell-to-cell variability in individual protein behavior. In summary, a gap remains in the image analysis field, requiring approaches that could (1) learn biologically meaningful features without human annotations, (2) produce universal features useful for studying subcellular organization at different scales, and (3) provide techniques for a wide range of downstream feature analyses.

Here, we present PIFiA (Protein Image-based Functional Annotation), a self-supervised approach for protein functional annotation derived from single-cell imaging data. PIFiA is coupled with a range of feature exploratory techniques for biological discovery. The representation learning component of PIFiA is performed by a convolutional neural network (CNN), which was trained with the objective of predicting protein identity directly from its fluorescently-labeled input image. This objective does not depend on pre-existing annotations or human labels and, unlike autoencoder-based models, PIFiA is robust to learning non-relevant information in the image, such as cell position, multiple cells in a crop, input noise, or imaging defects. In addition to the CNN component, the PIFiA workflow includes a set of downstream analysis steps for quantitative exploration of feature profiles extracted from single-cell imaging data. We applied PIFiA to ~3,000,000 live-cell confocal images of the budding yeast open reading frame (ORF)-GFP fusion collection (Huh et al, 2003). We compare PIFiA to existing approaches for protein representation learning and show that PIFiA outperforms previous methods on four different standards of protein function. We explore PIFiA feature profiles for use in a variety of downstream tasks, which are designed for the discovery of functional groups across different scales of cellular organization. Solely using distinct localization patterns of each protein, PIFiA can make remarkably precise functional predictions, identifying highly specific subcellular localization and distinct functional modules to reveal new biological insights.

# Results

## PIFiA architecture, feature profiles, and proteome-scale image dataset

PIFiA is a self-supervised deep learning approach designed to derive functional information about proteins from microscopy data without using any pre-existing annotations (Fig. 1). The PIFiA workflow consists of a feature extraction step performed by a deep neural network (Fig. 1A,B), as well as subsequent analysis steps on the extracted feature profiles (Fig. 1C–E). The downstream analysis enables prediction of protein localization and the identification of functional modules or subsets of proteins with related cellular roles, such as protein complexes and their associated regulators. The feature profiles can be used for multiple downstream tasks, including construction of a hierarchical map of subcellular organization (Fig. 1C), predicting protein function (Fig. 1D), identifying localization heterogeneity at a cell population level (Fig. 1E), and finding functional modules.

The deep learning backbone of PIFiA is a CNN consisting of eight convolutional blocks and three fully-connected (FC) layers,

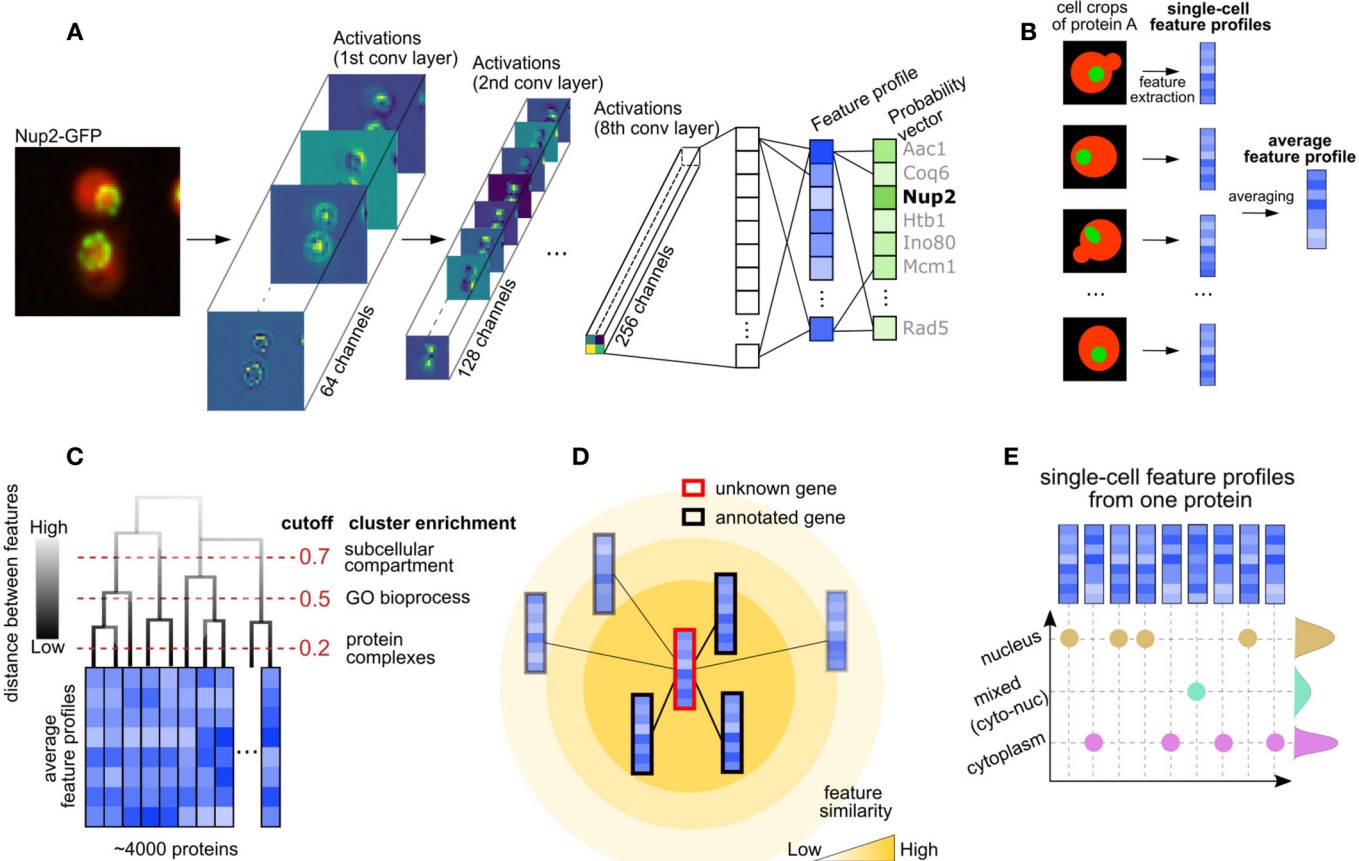

**Figure 1. Overview of the PIFiA workflow (see also Fig. EV1).**

(A) Diagram of the PIFiA neural network architecture. Shown are examples of activations from passing a micrograph of fluorescently labeled Nup2 protein (Nup2-GFP) through the PIFiA network, with corresponding patterns recognized by the convolutional filters. Feature profiles are extracted from the second fully-connected layer, for use in downstream applications. (B) Illustration of two types of feature profiles produced by PIFiA—single-cell feature profiles (extracted from a single crop) and averaged feature profiles (obtained by averaging all single-cell feature profiles of that protein). (C) Schematic representation of the global hierarchy of protein feature profile similarities to reveal different levels of functional information. (D) Illustration of protein function prediction using self-supervised PIFiA feature profiles. (E) An illustrative example of using PIFiA single-cell feature profiles to investigate the localization heterogeneity of a protein.

which was trained to predict a protein identifier associated with an input image (i.e. one out of 4049 classes (Fig. 1A)). The CNN produces a feature profile (or a representation profile) from the input image, which is unique to a particular image. Feature profiles are 64-dimensional real-valued vectors extracted from the second FC layer, which is followed by a classification layer (Fig. EV1A). These feature profiles encapsulate condensed information about each protein's identity, based solely on its localization pattern. Over the course of training, the model first learns straightforward characteristics, such as patterns of different cellular compartments, then it subsequently learns more subtle morphological features that may distinguish individual proteins (Fig. EV1B). To achieve the best accuracy and simplicity trade-off, we searched for the optimal architecture, network depth/width and related hyperparameters based on the validation set (Fig. EV1C,D) (see Methods). We found that more complex architectures, such as DenseNets (Huang et al, 2017), did not improve performance but increased the training time, hence we chose a simpler architecture that could achieve comparable performance. Similarly, we searched for optimal feature profile dimensionality and found that accuracy of protein identity

prediction stabilized around a 64-dimensional feature profile (Fig. EV1D).

To train PIFiA, we produced a comprehensive dataset of 3,058,961 live-cell images of individual strains expressing both a unique fusion gene from the yeast ORF-GFP collection (Huh et al, 2003) and spatial markers of cell cycle position (Huh et al, 2003), which provide cellular context for computational analysis of protein localization. In particular, we used automated yeast genetics (Tong and Boone, 2006) to engineer a new version of the ORF-GFP collection, in which the resultant strains also carried fluorescent markers of the nucleus (td-Tomato-NLS) and cytoplasm (E2-Crimson). In total, images of 4049 unique strains were obtained using an automated confocal microscope. Cell images were derived from two biological replicates, each of which had four fields of view for each ORF-GFP strain. The images acquired for the GFP channel were cropped into 64 × 64 pixels crops (median of 778 crops per tagged protein, see Methods), and each crop contained at least one cell at its center. The crops for each GFP-tagged protein were then split into training, validation, and test subsets (8:1:1 ratios).

After CNN training was completed, we extracted feature profiles of the individual single-cell crops from the test set to produce single-cell feature profiles (scFPs) (Fig. 1B). We then averaged the scFPs for each protein to create its average feature profile (aFP) (Fig. 1B). An aFP and scFP for an individual protein have the same dimensions, but they describe different levels of information: scFPs encapsulate the localization pattern of a protein in one cell, while aFPs describe the general spatial distribution of a protein. Below, we first use PIFiA aFPs to broadly explore protein localization and function. We then use scFPs to explore cell-to-cell heterogeneity, localization changes, and complex protein localization patterns (Fig. 1C–E).

## Comparison of PIFiA performance to other self-supervised and supervised approaches

We compared aFPs produced by PIFiA to the representations from three self-supervised methods (Fig. 2A–C) and two supervised methods (Fig. 2D,E). For self-supervised methods, we used CellProfiler (McQuin et al, 2018) (a feature-extraction tool) and its variant CellProfiler+PCA (see Methods), Paired Cell Inpainting (Lu et al, 2019) (a self-supervised autoencoder-based approach) and *cytoself* (Kobayashi et al, 2022) (a novel self-supervised method based on a VQ-VAE, that has been used to analyze human cell images). We also included a randomly initialized PIFiA network to show a baseline with the untrained model. For supervised approaches, we used DeepLoc (Kraus et al, 2017) (a deep-learning model that has been used previously to analyze images of the yeast ORF-GFP collection) and a combination of DeepLoc+PIFiA. We tried both the original DeepLoc version, which was trained on a set of 21,882 cell crops with single-cell labels, as well as our adaptation of DeepLoc, or DeepLoc+PIFiA, which was trained with less accurate but plentiful protein-level labels (see Methods).

We consider a model to have good performance if protein pairs with higher correlation between their aFPs are more likely to be

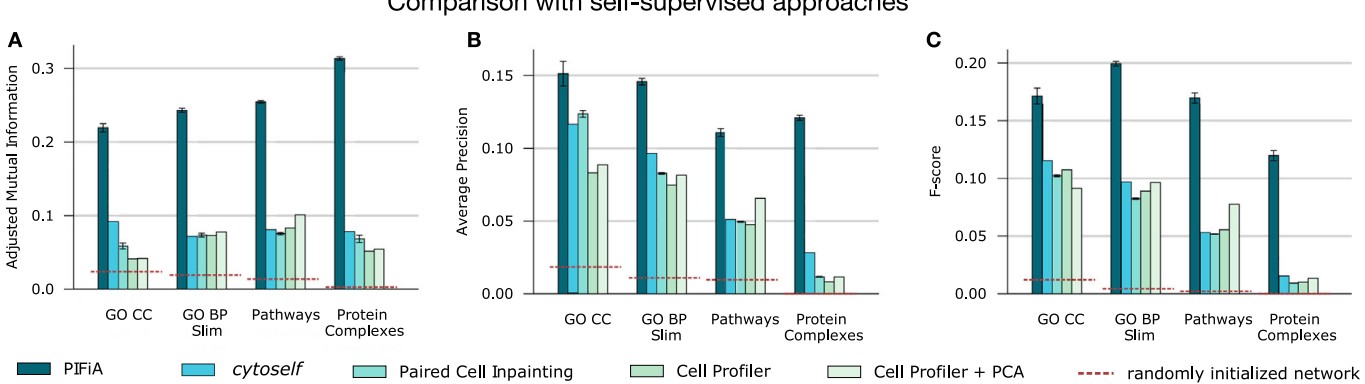

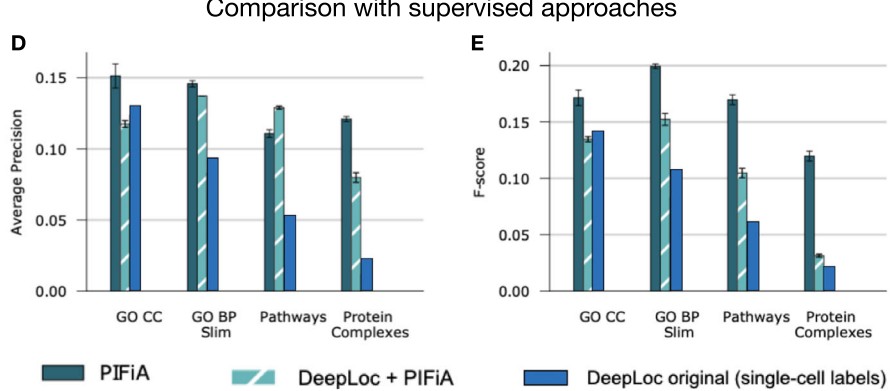

**Figure 2. Comparison of PIFiA performance to existing supervised and self-supervised methods for protein representation learning.**

(A) Bar graph showing the performance of PIFiA and four other methods (*cytoself* (Kobayashi et al, 2022), Paired Cell Inpainting (Lu et al, 2019), Cell Profiler (McQuin et al, 2018) and Cell Profiler + PCA) at detecting pairs of functionally related proteins (4049 total) using adjusted mutual information. A randomly initialized network (with PIFiA architecture) is shown for comparison as a dashed red line. Gene Ontology (GO) Cellular Component (CC), GO Slim Bioprocess (GO BP Slim), Kyoto Encyclopedia of Genes and Genomes Pathways (KEGG pathways) and European Bioinformatics Institute protein complexes (Protein Complexes). Error bars represent standard deviation from the mean across three network runs. (B) Bar graph (same setup as A) showing performance assessed using average precision. (C) Bar graph (same setup as A) showing performance assessed using F-score on four biological standards. (D) PIFiA performance versus supervised approaches (two variations of DeepLoc (Kraus et al, 2017)) assessed using average precision on four biological standards [X axis: Gene Ontology (GO) Cellular Component (CC), GO Slim Bioprocess (GO BP Slim), Kyoto Encyclopedia of Genes and Genomes Pathways (KEGG pathways) and European Bioinformatics Institute protein complexes (Protein Complexes). "DeepLoc original" is the original implementation (Kraus et al, 2017) trained on crops with manually annotated single-cell labels (4049 proteins total); DeepLoc+PIFiA is our modified version trained on protein-level labels using Huh et al. localization standard (Huh et al, 2003). Error bars indicate the standard deviation of the scores across three independent runs (for deep learning models). (E) Bar graph (same setup as E) showing performance assessed using F-score.

functionally related. We evaluated feature profiles (aFPs) using three metrics: F-score and average precision (AP), both measures of feature relevance, and adjusted mutual information (AMI), an information theoretic metric to assess clustering quality (see Methods). PIFiA features showed superior performance on most evaluation criteria using four functional benchmarks: Gene Ontology (Harris et al, 2004) (GO) Cellular Components (CC), GO Slim Bioprocesses (GO BP slim), Kyoto Encyclopedia of Genes and Genomes (Kanehisa and Goto, 2000) (KEGG) pathways and European Bioinformatics Institute (EBI) Protein Complexes (Meldal et al, 2015) (Fig. 2).

PIFiA reached better performance than the supervised method DeepLoc in predicting protein subcellular localization (DeepLoc's target task), as indicated by higher values of F- and AP scores, on the GO CC standard (Fig. 2). PIFiA also outperformed DeepLoc based on other functional standards, with the biggest performance gain in protein complex discovery. This result confirms the utility of the PIFiA training objective which targets identification of individual tagged proteins, the most detailed level of functional information present in the image. Although the objective does not directly focus on localization prediction, over the course of training the CNN implicitly learns a variety of localization patterns needed to successfully differentiate individual proteins. Thus, PIFiA self-supervised feature profiles can be used for exploratory analysis of protein localization instead of representations from a supervised method such as DeepLoc, bypassing the need for manual annotation while improving performance.

PIFiA also demonstrated better performance than Paired Cell Inpainting, another self-supervised method, achieving 1.2, 1.7, 2.2, and 10.4-fold improvements in terms of mean average precision using cellular component, bioprocess, pathway and protein complex standards, respectively. Also, PIFiA outperformed *cytoself*, a self-supervised approach that utilizes a VQ-VAE in its architecture, which achieved similar performance to Paired Cell Inpainting. Compared to all other approaches examined, PIFiA representations resulted in substantial improvement in clustering quality measured by AMI scores, with an average 5-fold AMI improvement over Paired Cell Inpainting (Fig. 2A). The significant improvement on the protein complex standard is again explained by PIFiA's novel self-supervised objective, which forces the network to detect the most comprehensive morphological patterns while ignoring individual image artifacts, which contrasts with autoencoder-based objectives that learn features by naive image reconstruction.

Finally, to evaluate PIFiA performance specifically on proteins that localize to compartments with similar morphologies, we did an extra evaluation run similar to Fig. 2A, but only including aFPs of proteins from Golgi, endosome and peroxisome (with a single Huh et al localization label). We obtained 0.13, 0.09, 0.08, and 0.1 AP scores for PIFiA, Paired Cell Inpainting, DeepLoc original and DeepLoc+PIFiA, confirming that PIFiA is capable of distinguishing proteins from compartments with similar morphologies.

## Evaluation of the functional information associated with PIFiA average feature profiles

To further assess the biological information associated with the aFPs of each protein, we used hierarchical clustering of aFPs as an unsupervised approach to discover feature profile similarities (Murtagh and Contreras, 2012). We performed agglomerative

hierarchical clustering of the whole-proteome aFPs (4049 proteins in total) using a correlation metric and average linkage. We surveyed the resulting dendrogram at different thresholds to explore whether aFPs are suitable for studying the spatial architecture of the cell at different scales of its organization (Fig. 3A; Appendix Fig. S1). The hierarchical clustering results are shown in Fig. 3A, with 4049 proteins on the X-axis clustered according to the similarity of their feature profiles (each column is a 64-dimensional aFP). To determine optimal cutoff thresholds, we tracked AMI scores (Vinh et al, 2010) at different correlation thresholds for three functional standards: GO Cellular Component, GO Slim Bioprocess and Protein Complexes (Appendix Fig. S1B) (see Methods).

First, we determined an optimal threshold (0.72) corresponding to the most general level of cellular organization—GO Cellular Component annotations (Fig. 3A). The nine clusters produced at this threshold were enriched for proteins with relatively broad cell component annotations: nucleus, mitochondrion, Golgi apparatus, cytoplasm, endoplasmic reticulum (ER), actin patches, nucleolus and cytosolic ribosome (all $p$-value $< 10e-20$ except cluster 4 with $p$-value $< 10e-5$; Fig. 3A,B; Dataset EV1; examples of cell images from each cluster are shown in Appendix Fig. S1A). At this level of feature profile similarity, proteins annotated to subcellular components with visually distinct morphologies, such as organelles, tend to be in a single cluster, whereas proteins annotated to more heterogeneous cellular compartments are found in multiple clusters. For example, proteins with a *nucleus* GO cellular component annotation are enriched only in cluster 1, whereas proteins with a *cytoplasm* annotation were enriched in clusters 4, 8, and 9 (Fig. 3A,B). Detailed visual image inspection revealed that some clusters reflect protein localization to both the cytoplasm and another compartment, such as cluster 4 which contains subsets of proteins localized to the cytoplasm and cell surface proteins. Other clusters likely reflect differences in protein abundance, such as cluster 8, which includes a number of highly abundant proteins, including ribosomal proteins.

Next, we derived optimal correlation thresholds on our dendro-gram corresponding to two additional, more detailed biological standards: GO Slim Bioprocess and Protein Complexes (Appendix Fig. S1B). We obtained 21 clusters for GO Slim Bioprocesses (0.64 AMI distance threshold), 20 of which were functionally enriched (GO enrichments are shown in Appendix Fig. S1C; median $p$-value of $5e-10$ across all enriched clusters; cluster entropies in terms of present localizations are shown in Appendix Fig. S1D). Similarly, 205 clusters were found at the dendrogram cutoff corresponding to a protein complex standard (0.29 AMI, Appendix Fig. S1B), which had 11-fold median enrichment in protein complex predictions across all clusters (distribution of the per-cluster enrichments at 0.29 AMI cutoff is shown in Appendix Fig. S1E,F). Hence, aFPs present robust and memory-efficient representations of protein features, which allow detection of clusters with functionally related proteins at various levels of cellular organization, with the highest functional resolution at more general levels of the hierarchy (Appendix Fig. S1F).

## Adaptation of PIFiA features to external annotations for protein localization and function

Our clustering analysis showed that PIFiA aFPs capture information from cell images that enables unsupervised resolution of cellular spatial

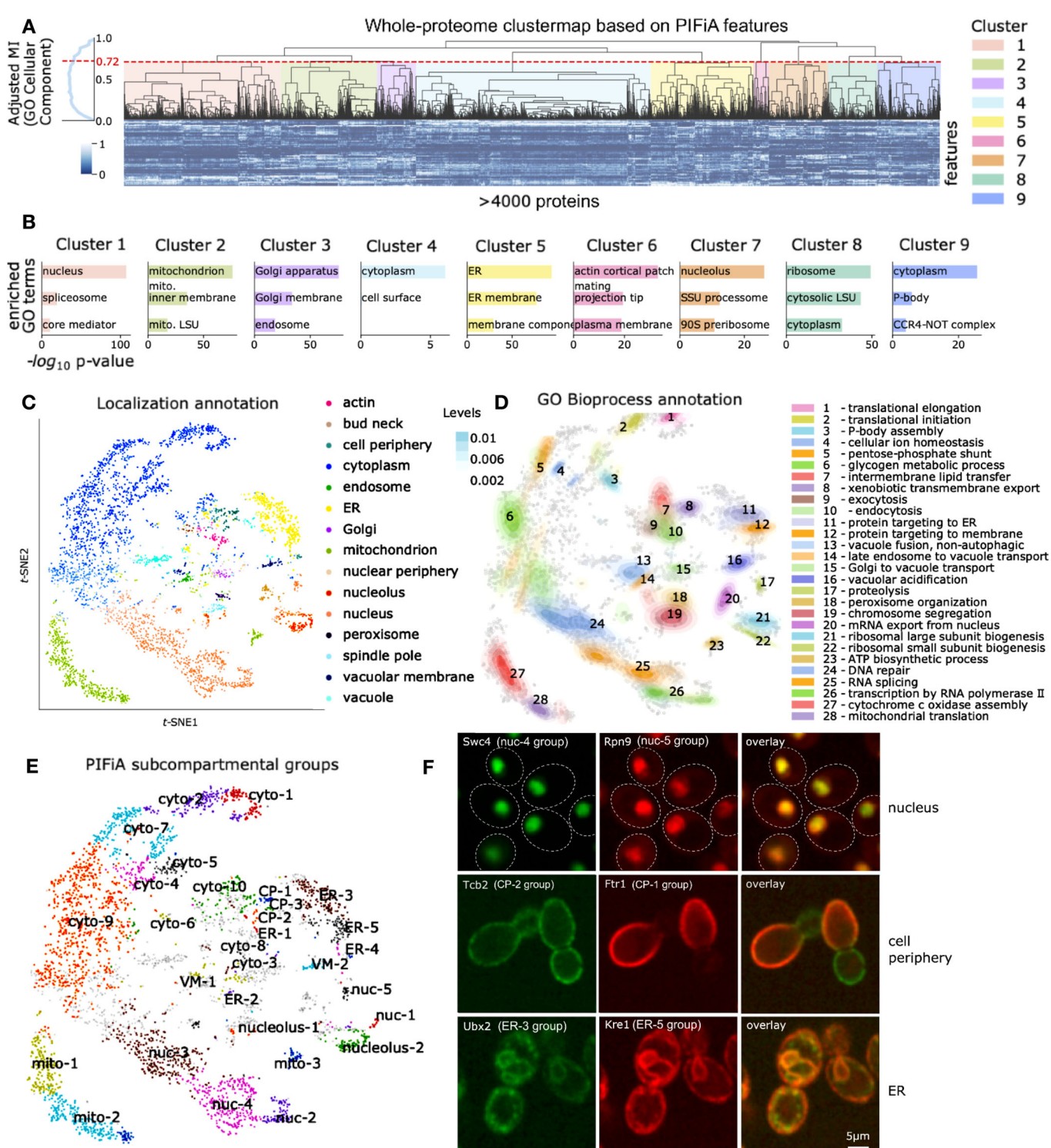

organization, grouping proteins by shared localization and biological function (Fig. 3A,B). We have investigated another useful property of feature profiles—adaptability for subsequent supervised training. One of our goals was to create a model that produces universal feature profiles that can be used without the requirement to re-train a full neural network from scratch on a specific task. To evaluate the adaptability of feature profiles, we used the widely adopted linear evaluation protocol (Chen et al, 2020) where a linear classifier is trained on top of the representations extracted from the network, and test accuracy is used as a measure of representation quality.

We first evaluated how PIFiA features can be adapted to protein localization labels, which comprise the largest labeled functional standard available (Huh et al, 2003; Kraus et al, 2017). This analysis enables assessment of whether information contained

**Figure 3.   Clustering of PIFiA average feature profiles and analysis of the associated biological information (see also Figs. EV2, EV3, EV4, EV5; Dataset EV1 and EV2; Appendix Fig. S1).**

(A) Clustergram of PIFiA's average feature profiles for 4049 proteins. The plot to the left of the Y axis shows the adjusted mutual information curve between clustering labels and GO Cellular Component labels at different distance thresholds. The distance threshold ($d = 0.72$) indicated on the clustergram produces clusters associated with cell compartments (color codes on the right). (B) Bar graph showing top three Gene Ontology Cellular Component scores for each cluster defined in (A). (C) Whole-proteome tSNE projection of PIFiA average feature profiles. Each point on the plot represents a protein colored according to a localization category predicted by logistic regression (see Results for training details). (D) Annotation of the whole-proteome tSNE projection with GO bioprocess categories shown as Gaussian kernel density estimates. Bioprocesses were selected according to the lowest variance from different cellular components. A filled contour plot was used instead of contour lines to make bioprocess groups more easily distinguishable. The color intensity of the kernel density estimate contour plots corresponds to the cumulative probability mass below the drawn contour. (E) Annotation of the whole-proteome tSNE projection with sub-compartmental protein groups predicted by clustering of PIFiA feature profiles (cyto—cytoplasmic cluster, nuc—nuclear cluster, mito—mitochondrial cluster). (F) Colocalization analysis of proteins from different sub-compartmental groups. Representative micrographs of cells expressing mNeonGreen- (left) or mScarlet- (middle) tagged proteins annotated to different sub-compartmental groups within three cellular compartments: *nucleus* (top), *cell periphery* (middle) and *endoplasmic reticulum* (ER, bottom) groups. Overlays of the mNeonGreen and mScarlet images are shown on the right. The tagged proteins are indicated on the micrographs.

in self-supervised PIFiA features matches the content of the original images, when extracted with a supervised method. We trained a multinomial logistic regression (LR) using PIFiA scFPs from 2415 proteins manually annotated to localize to a single subcellular localization (Huh et al, 2003) (see Methods). We compared the final performance of the LR trained on PIFiA scFPs to DeepLoc, a supervised neural network specifically trained to classify protein localization from images of the yeast ORF-GFP collection. To match the training protocol of DeepLoc, we used scFPs derived from the single-cell images in DeepLoc's training set (Kraus et al, 2017). We report AP scores on the same single-cell crops from the test set across the full roster of 2415 single-localized proteins (Fig. EV2A). Remarkably, PIFiA self-supervised scFPs that were paired with LR yielded a comparable performance to the supervised network DeepLoc, even though PIFiA feature profiles are self-supervised and were fitted to localization labels solely using LR. This finding suggests that PIFiA feature profiles have rich functional content, and we can use them to predict functional protein attributes without training a full network from scratch.

To visualize adaptation of PIFiA feature profiles to the supervised localization labels, we transformed the 64-dimensional aFPs into 2-dimensional space using t-SNE (Van der Maaten and Hinton, 2008) and colored them according to LR localization predictions (Fig. 3C). Each aFP on the t-SNE projection was annotated with the localization category corresponding to the maximal LR prediction across all scFPs. In this visualization, the morphological similarity of proteins encapsulated in the aFPs was translated into proximity on the 2D t-SNE map, highlighting that the separation of self-supervised aFPs on the map was driven by subcellular localization signals. We compared the aFP localization assignments with the assignments made using supervised machine learning methods or manual annotations trained to specifically assign proteins to subcellular localizations (Fig. EV2B). Ultimately, we observed higher quality of linear localization annotation compared to subcellular localization standards produced by other approaches (Chong et al, 2015; Kraus et al, 2017) (Fig. EV2A).

These analyses show that PIFiA feature profiles can be adapted to the objective of a supervised neural network, which confirms the high information content of PIFiA features. Such adaptable feature profiles may accelerate training by replacing various task-specific supervised neural networks with one multi-purpose self-supervised approach, which yields universally applicable representations.

## Generalization to unseen datasets

We investigated the generalization capabilities of the PIFiA neural network by applying it to previously unseen yeast imaging datasets. Specifically, we evaluated the performance of PIFiA on the publicly available CYCLoPS (Chong et al, 2015) and YeastRGB (Dubreuil et al, 2019) datasets. Both datasets contain images of >4000 unique GFP-tagged proteins: CYCLoPs images show a version of the ORF-GFP collection and YeastRGB contains images of a new collection based on a different fluorescent protein, mNeonGreen. We applied the PIFiA network out-of-the-box to single-cell crops of the fluorescently-tagged protein images from these datasets and extracted their feature profiles (see Methods for details on feature extraction).

We generated aFPs from both datasets and used tSNE to visually represent the similarity between feature profiles, with points in close proximity reflecting similar visual characteristics (Fig. EV3). We color-coded the tSNE maps according to the annotated subcellular localizations of the yeast cell components using a well-known standard of yeast protein localization (Huh et al, 2003). This visualization strategy allowed us to observe the formation of dense clusters corresponding to specific subcellular compartments. Nucleus, cytoplasm, mitochondrion, ER, vacuole and nucleolus were the most distinguishable localizations across both datasets (Fig. EV3A,B). Thus, the PIFiA network showed strong generalization capabilities on two unseen datasets without any fine-tuning of its weights. We attribute this generalization to diverse data augmentation that was applied to the training data. Overall, our results confirm the feasibility of applying PIFiA for the analysis of novel datasets.

## Experimental validation of PIFiA predictions of sub-compartmental organization of the cell

We explored more specific functional information associated with PIFiA aFPs. We used Gaussian kernel density estimates (KDEs) (see Methods) to annotate our whole-proteome 2D tSNE projection of aFPs with Gene Ontology bioprocess terms. For illustration, we selected terms from different subcellular components that had the lowest variance on the t-SNE map. This annotation showed that PIFiA features distinguished biological processes within cellular compartments (Fig. 3D). For example, regions of the tSNE map corresponding to the cytoplasm (Fig. 3C) had distinct regions enriched for translation initiation and elongation, P-body assembly,

pentose-phosphate shunt and glycogen metabolic process (Fig. 3D). This analysis illustrates that GFP-tagged proteins with similar biological roles have distinguishable appearances in cell images, and that PIFiA learns feature profiles that can be used to discover protein functional groups across different levels of subcellular organization, including organelles and possible sub-compartmental structures.

To further explore information in PIFiA profiles related to the sub-compartmental organization of the cell, we clustered aFPs of proteins that mapped to the same localization category to produce 15 per-compartment hierarchical trees (derived from the 15 categories defined by LR; Fig. 3C). We selected a sub-compartmental clustering threshold of 0.5 based on the highest morphological similarity within clusters and maximal separation between clusters, measured by a Silhouette score (Appendix Fig. S1E). We identified 30 clusters, which are indicated on the tSNE projection of PIFiA feature profiles in Fig. 3E, with example images of cells from each group shown in Fig. EV4 (see also Dataset EV2 for GO enrichment and other information). We refer to these clusters using their localization category and associated group number (e.g., nuc-1 corresponds to the first sub-compartmental group in the nucleus). We provide an interactive version of the t-SNE plot from Fig. 3E on the PIFiA website (https://thecellvision.org/pifia/), where each point on the plot is clickable and allows the user to explore the micrographs corresponding to the GFP-tagged protein, find its nearest neighbors and perform enrichment analysis based on the closest aFPs.

Several general features associated with the clusters in Fig. 3E suggest that they are functionally meaningful and reflect sub-compartmental organization. First, proteins localizing to compartments which tend to be more homogeneous in their morphological patterns were typically seen in a single cluster (e.g., peroxisome, spindle pole, vacuolar membrane, nuclear periphery), while proteins associated with large or heterogeneous compartments, such as the nucleus, cytoplasm, and mitochondria, defined more than one sub-compartmental cluster (Fig. 3E; Dataset EV2). Second, 16 of 32 groups showed >2-fold enrichment for a GO annotation category (Dataset EV2, $P < 0.01$). For example, the nucleus region of the whole-proteome map was divided into five clusters, enriched in GO bioprocesses such as small molecule metabolic process, chromatin remodeling and RNA polymerase II activity, mitotic nuclear division and proteolysis (Fig. 3E; Dataset EV2). Third, as expected, some of the groupings appeared to be based on protein features that were easily discernible. Proteins in some sub-compartmental groups have a tight distribution of GFP intensities, suggesting that abundance is likely an important feature for that group. For example, nuc-1 clustering likely resulted from high protein abundance, and this cluster included histones and metabolic enzymes (median GFP intensity nuc-1 proteins = $5834 \pm 2103$ vs median for all nuc proteins = $745 \pm 793$) (Dataset EV2). Likewise, nuc-3 proteins had low abundance (median GFP intensity = $678 \pm 52$) and this group was enriched for mitotic nuclear division and chromosome segregation. For some of the other groups, clustering appeared to result from differences in the spatial distribution of pixels in a region. For example, cyto-3 proteins all had a prominent cytosolic signal overlaid with a punctate morphology, and most had roles in Golgi vesicle transport (Dataset EV2; Fig. EV3). Similarly, cyto-8 contained only seven proteins with no obvious functional overlap, but by visual inspection, all the proteins localized to the cytoplasm and to one or more foci (Fig. EV3). Thus, a fraction of sub-compartmental groups could be explained by distinct protein localization features, which may correspond to coherent functionality.

For many sub-compartmental groups, however, the features driving the clustering were less obvious. To ask if we could manually identify differences between proteins from different PIFiA sub-compartments with the same overall localization, we used a more sensitive colocalization assay. We chose pairs of proteins with similar abundances, tagged them with two fluorescent proteins, mNeonGreen and mScarlet, imaged cells containing both tagged proteins, taking Z-stacks of 5 optical sections, and manually assessed images (Fig. 3F; Dataset EV2). Using colocalization, we observed subtle differences in most of the pairs from different sub-compartmental groups; specifically, we identified differences in 39/52 (75%) protein pairs from distinct groups, but in only 7/24 (29%) pairs from the same group (Dataset EV2). We show three examples of pairs of proteins from different sub-compartmental groups that vary in their localizations to different extents. In one example with a clear difference, we identified a distinct localization for nuc-5 proteins, which were 13.5-fold enriched for components of the proteasome ($P = 7.78E-22$). The localization of nuc-5 proteins overlapped extensively with that of other nuclear proteins, but nuc-5 proteins additionally localized to the nuclear periphery (Fig. 3F, top row). We detected the nuclear periphery localization of nuc-5 proteins when we looked at different proteasome components from nuc-5, in colocalization assays with proteins from different sub-compartmental nuc groups, and when the fluorescent proteins were reversed (Fig. EV5).

In another example with a clear difference, we performed co-localization analysis with proteins assigned to different cell periphery (CP) groups. We examined cells expressing both a high-affinity iron transporter, Ftr1, from the CP-1 group, and Tcb2, a protein involved in ER-plasma membrane tethering, from the CP-2 group (Fig. 3F, middle row). Ftr1, tagged with mScarlet, and Tcb2, tagged with mNeonGreen, localized to distinct regions of the cell periphery. Ftr1 localized specifically to the mother cell periphery but was absent from the bud, whereas Tcb2 was present at both the mother and daughter cell peripheries. Indeed, by visual inspection, we found that many of the CP-1 proteins had apparent mother-specific localization, like Ftr1. In total, the CP-1 group contains 21 proteins, and includes 7 of the 8 proteins found previously to localize asymmetrically to mother cells, all of which are members of the MDR (multidrug resistance) transporter family: Fui1, Hip1, Hnm1, Pdr5, Snq2, Tpo1, Yor1 (Decottignies et al, 1998; Eldakak et al, 2010).

In addition to the MDR transporters, the CP-1 group contains 14 novel mother-specific proteins, including several other transporter proteins (Atr1, Ftr1, Hxt6, Mep1, Mep3, Qdr2, and Qdr3), proteins with roles in signaling (Gpa2, Mid2, Psr1), and three relatively uncharacterized proteins (Ybr016w, Ina1, and Crp1; Dataset EV2).

In a third example, we looked at colocalization of two proteins whose ORF-GFP fusions show some ER localization: Ubx2, a protein involved in ER-associated protein degradation from the ER-3 sub-compartmental group (Neuber et al, 2005), and Kre1, a protein that normally functions as a cell wall glycoprotein from the ER-5 group (Boone et al, 1990). The difference between these is more subtle: the ER-5 protein, Kre1, shows an ER localization but

also an increased concentration at the cell periphery compared to Ubx2 (Fig. 3F, bottom row). This localization difference was observed in other members of these sub-compartmental groups, with *ER-3* proteins tending to have a more diffuse localization and *ER-5* proteins localizing more to the cell periphery (Fig. EV5).

In summary, our data show that within a compartment, PIFiA features can distinguish groups of proteins with subtle differences in localization that often have different biological roles. Many of these groups are enriched for proteins that perform biological functions not previously associated with distinctive localization patterns.

## Analysis of proteins with multi-compartment localization using PIFiA single-cell feature profiles

The single-cell feature profiles (scFPs) produced by the PIFiA CNN provide an opportunity to explore more nuanced protein behaviors, including proteins localizing across multiple compartments. Previous analyses of the yeast ORF-GFP collection showed that a large fraction of the proteome localizes to two or more compartments in the same cell (Chong et al, 2015; Huh et al, 2003; Kraus et al, 2017). These studies used average statistics for cell populations, precluding differentiation of proteins that localize to multiple compartments, or those that shuttle between compartments. We annotated scFPs of every protein with localization categories using LR classification scores, and then we investigated the distribution of each protein's single-cell localization scores, focusing on the two most probable localizations (see Methods). Using this strategy, we found that most (3424) proteins have a homogeneous localization (localizing to a single compartment), while 652 proteins exhibit localization heterogeneity (localizing to two or more compartments) (Fig. 4A). We classified the proteins with heterogeneous localization into two categories: (1) 396 proteins that localized to more than one compartment in a single cell, which we refer to as AND-proteins, and (2) 256 proteins that appeared either in a primary or a secondary location but not in the same cell, which we refer to as OR-proteins (Fig. 4B,C,D; Dataset EV3). For most proteins the assigned localization probabilities were continuously distributed, but our classification summarizes the localization, indicating the compartments that the protein predominantly populates. For example, Pho85 was classified as an AND-protein with a mixed signal predominantly from nucleus and cytoplasm within single cells, consistent with its known biology (Huang et al, 2007) (Fig. 4C). In contrast, Stb1 is a transcription factor whose nuclear localization is cell cycle regulated and it was classified by our analysis of scFPs as being either nuclear or cytoplasmic (OR-protein), as seen in previous studies (Youn et al, 2017) (Fig. 4C).

We summarized overall AND-/OR-localizations across 15 localization categories, which clearly illustrated that a large fraction of these changes involved the nucleus and cytoplasm compartments. Among the 922 proteins with a high confidence nuclear localization, 708 were solely nuclear, 159 nuclear AND cytoplasmic, and 55 nuclear OR cytoplasmic (Fig. 4E,F). We asked how these classes were distributed in different bioprocesses involving the nucleus (Costanzo et al, 2016). As expected, proteins with roles in RNA processing and chromatin organization tended to be solely nuclear (Dataset EV3). The trends for proteins that have dual localization were also consistent with well-established biology. For example, proteins with roles in cell cycle progression were 4.1-fold enriched in nucleus OR cytoplasm ($P = 6.7E-05$). Many cell cycle

proteins, in particular many transcription factors, use localization to regulate protein activity (Haase and Wittenberg, 2014). Proteins with roles in DNA replication/repair and stress response were weakly enriched in nucleus AND cytoplasm (1.5-fold, $P = 1.40E-03$ and 1.9-fold, $P = 9.7E-04$, respectively). DNA repair proteins often alter their relative localization in the presence of damage, either to initiate a repair response or to prevent catastrophic cell cycle events (Tkach et al, 2012). Because our cells were not experiencing DNA damage at the time of imaging, many of these proteins displayed both nuclear AND cytoplasmic localization in our data. Hence, while many protein groups that show different patterns were too small to perform consistent enrichment analysis, enrichments that were seen for nuclear-cytoplasmic groups, where there are enough proteins to assess, were consistent with known biology.

Finally, because proteins with roles in cell cycle progression were enriched among both the OR- and the AND-proteins, we used our scFPs to assess how cell cycle position could account for some of the protein localization heterogeneity. To do so, we took advantage of the nuclear and cytoplasmic markers (td-Tomato-NLS; E2-Crimson) in our GFP collection to explore the relationship between cell cycle position and protein abundance or localization heterogeneity. We first trained an ensemble of CNNs on the nuclear and cytoplasmic RFP channels to predict one of three cell cycle stages, and we subsequently mapped each single-cell crop to a cell cycle category: T/G1 (Telophase, Gap phase 1), S/G2 (DNA synthesis phase/Gap phase 2) or M/A (metaphase/anaphase) (see Methods). We then compared the single-cell distributions of each cell cycle stage with the localization calls to discover relationships between protein localization and cell cycle (Dataset EV3; Appendix Fig. S2). In total, we identified 136 proteins with cell-cycle-dependent variation in PIFiA feature profiles, determined by Mann–Whitney U test (McKnight and Najab, 2010) (*p*-value < 1e−3, see Methods). Our results are summarized in the connected heatmap shown in Fig. 4G. As expected, some of the discovered localization changes reflected cell-cycle-dependent differences in the corresponding compartment. For example, most bud neck/cytoplasm AND-localizing proteins (14/23 proteins) were cytosolic in G1 before the bud neck had formed and localized to the bud neck later in the cell cycle (Dataset EV3). However, many cell cycle-regulated proteins moved between permanent compartments, including 66 moving between the nucleus and cytoplasm. Indeed, PIFiA identified 4 proteins not previously known to be cell cycle regulated, that localized to the cytoplasm and nucleus, but showed a predominantly cytoplasmic localization in M/A (Yel025c, Atc1, Bop3, and Cmg1).

Overall, scFPs derived from the self-supervised PIFiA workflow enable resolution of single-cell localization and are suitable for cell-to-cell variability analysis. Notably, PIFiA feature profiles contain enough functional information to distinguish compartments and sub-compartmental morphologies without pre-assigned labels, enabling analysis of protein localization heterogeneity in a data-driven way, which precludes propagating annotation errors.

## Prediction of functional modules using PIFiA single-cell feature profiles

We showed that protein-level feature profiles, or aFPs, can present a range of microscopy patterns in a compressed numerical form,

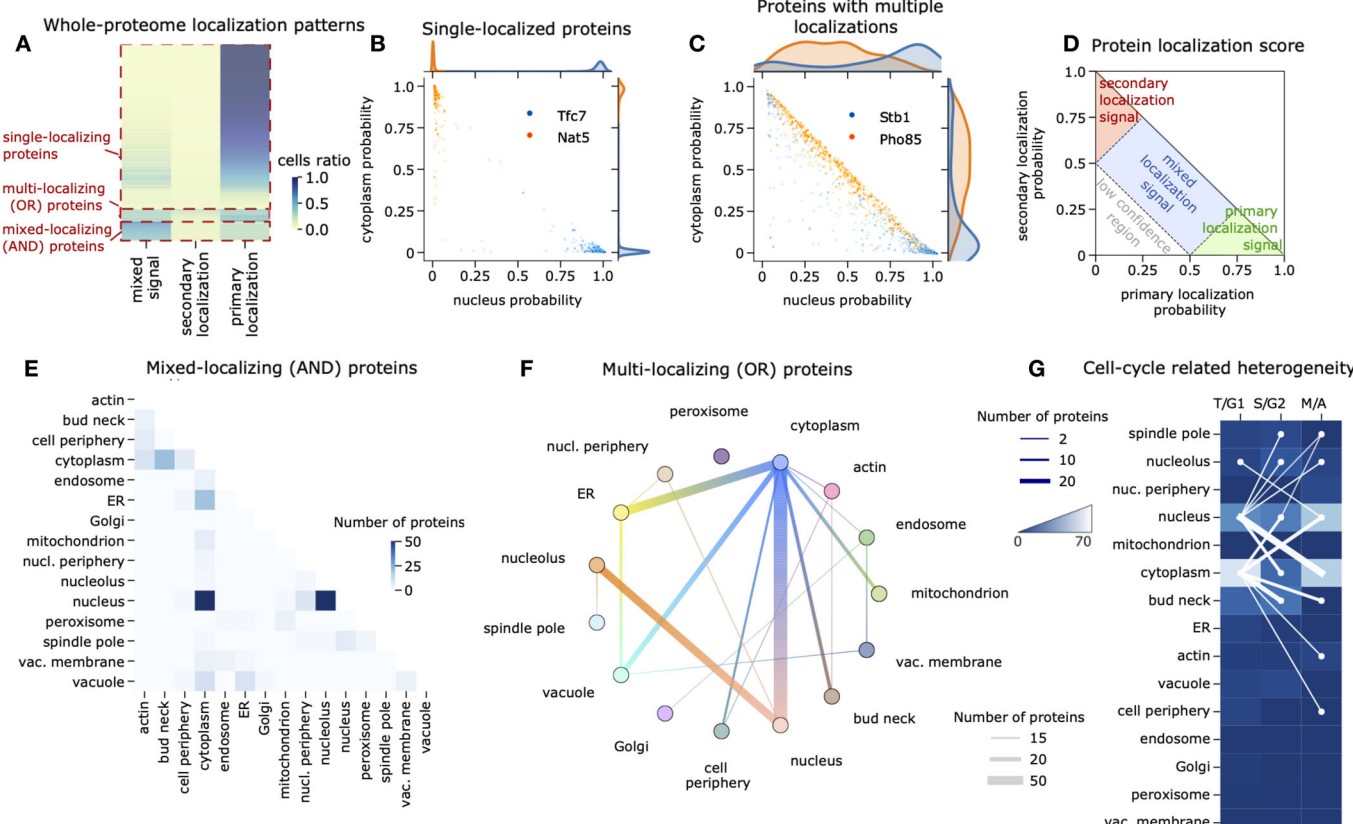

**Figure 4. Identification of proteins with morphological heterogeneity using PIFiA single-cell feature profiles (see also Dataset EV3; Appendix Fig. S2).**

(A) Heatmap depicting ratios of cells falling into a mixed localization category, secondary and primary localization regions. Proteins with low ratios in all three columns (yellow color) feature many cells that fall into the low-confidence region. (B) Scatter plot showing two proteins with homogeneous localization patterns. Each point on the scatter plot corresponds to a single-cell crop, mapped to the probability of nuclear and cytoplasmic localization according to the LR predictions. (C) Scatter plot showing a protein with AND-type localization heterogeneity, Pho85, and one with OR-type localization heterogeneity, Stb1. (D) Schema of scoring proteins for localization heterogeneity using a single-cell level distribution of localization probabilities. Probabilities are obtained from primary and secondary localizations (i.e. first and second most probable localizations of the logistic regression classification of that protein). (E) Localization co-occurrence heatmap for 396 AND-localizing proteins, showing numbers of proteins present at two localizations. The scale bar is set to a maximum intensity of 50 to enable visualization of categories with fewer proteins. (F) Circle plot depicting localization patterns of 256 OR-type proteins. The thickness of the line connecting two localizations indicates the number of proteins showing localization heterogeneity between these localizations. (G) Localization heterogeneity related to cell cycle position for 136 proteins exhibiting statistically significant cell cycle variation. Connections indicate localizations of the proteins at specific cell cycle stages (thicker lines indicate a more common connection between a particular localization change and cell cycle stage transition). The color of the heatmap indicates the number of heterogeneous proteins that are present in the corresponding cell cycle phase for a particular localization.

which can be used for clustering and building hierarchical dendrograms. Using AMI scores at different correlation thresholds we were able to resolve functional information associated with hierarchical clustering of PIFiA aFPs (Fig. 2A) and determine an optimal correlation threshold for discovering functional modules, such as protein complexes (Appendix Fig. S1A). However, averaging feature profiles leads to information loss, which is not optimal for more precise analysis. Hence, we explored the use of single-cell feature profiles for the identification of functional modules. In particular, we focused on whether we could use scFPs for improved identification of protein complexes, which represent functional modules whose components are expected to colocalize within a single cell.

We derived our scFPs clustering analysis from a straightforward intuition—scFPs belonging to the same protein or the same protein complex should be indistinguishable, given the resolution limits of

light microscopy. To visually illustrate this hypothesis on protein complex member distributions with scFPs, we projected scFPs from the test set using 2D tSNE (Fig. 5A). The scFPs of proteins from the same complex often localized together on the tSNE map, but different proteins from the same complex were typically intermingled and difficult to separate from each other. In contrast, the scFPs corresponding to different protein complexes with the same subcellular localization were often separated on the tSNE map (e.g., Polymerase-II, Polymerase-III, and RSC in the nucleus; EGO and V-ATPase in the vacuolar membrane) (Fig. 5A).

To further explore the utility of scFPs for algorithmically identifying protein complexes, we developed a modified hierarchical clustering approach, called adaptive thresholding, that is designed to identify correlation thresholds on the hierarchical dendrogram at which scFPs inside a cluster become indistinguishable, and thus might be expected to contain interacting proteins

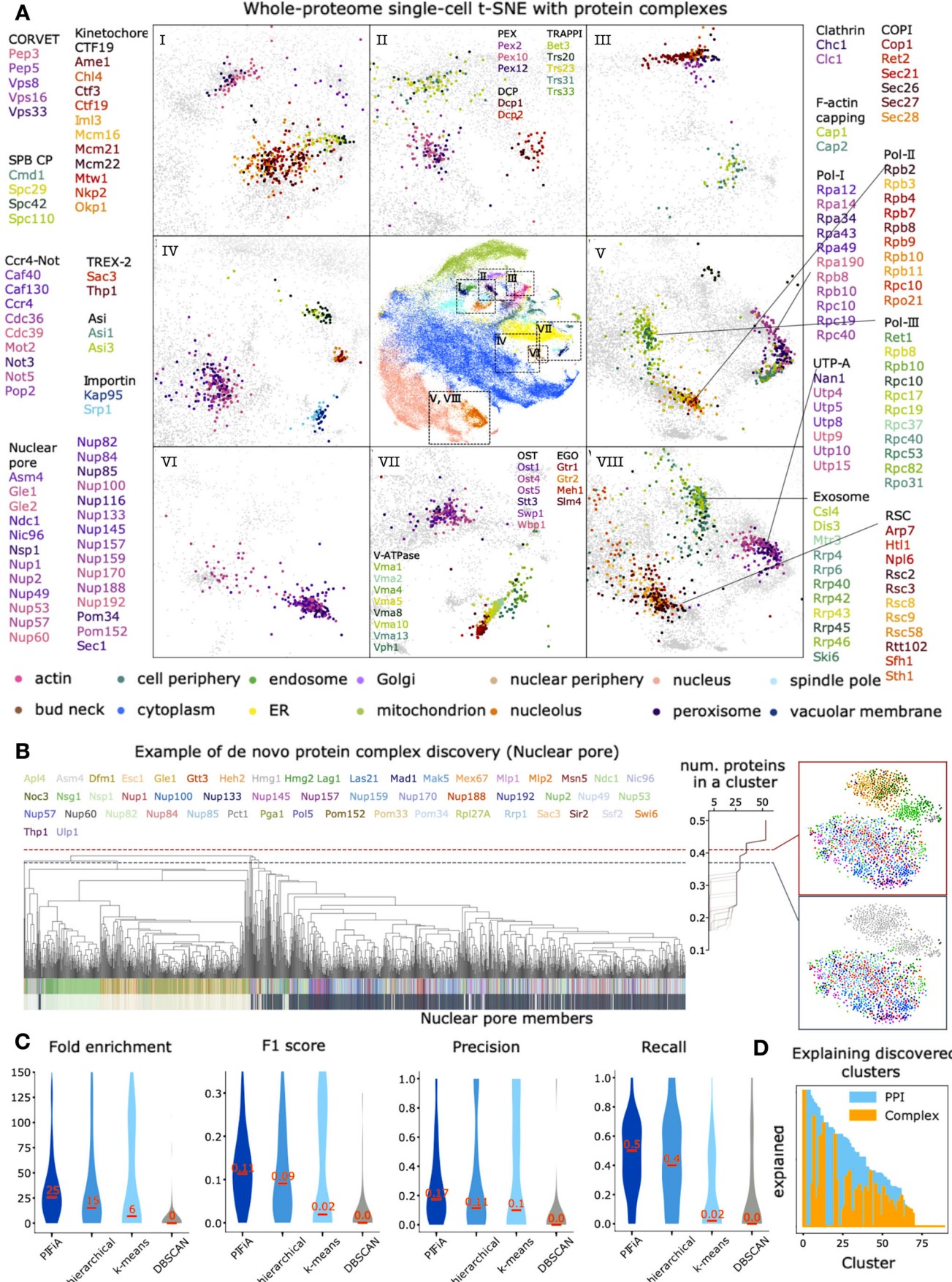

Figure 5.  Prediction of protein functional modules using PIFiA single-cell feature profiles (see also Dataset EV4).

(A) Visualization of protein complex clusters on a single-cell tSNE plot of PIFiA feature profiles. The central plot shows a whole-proteome tSNE projection of PIFiA single-cell feature profiles (scFPs). Each point on the plot represents a protein that is colored according to 15 different subcellular localizations (color codes are explained below the plot). Zoom-in plots show a more detailed view of some regions of the global tSNE, showing single-cell features from the test set corresponding to proteins from the same complex with the same color palette, each protein shown in different color. (B) Dendrogram of clustered scFPs highlighting a region that identifies a nuclear pore cluster among nuclear periphery scFPs. The line graph on the right shows different numbers of proteins in a cluster at different correlation thresholds for clustering. Zoom-in plots of two clusters at different correlation thresholds (red and gray dashed lines) are shown as scFPs tSNE plots to the right. (C) Violin plot comparing the performance of four clustering approaches on 140 protein complexes. (D) Plot illustrating the fraction of proteins in each cluster with a protein–protein interaction annotated in the BioGrid (blue) or as protein complex member (yellow). Cluster numbers are sorted based on the number of discovered interactions.

(see Methods). We performed hierarchical clustering of test set scFPs using average linkage and a correlation metric. We then traced the number of unique proteins inside the cluster along with the divisions of the single-cell dendrogram to discover levels of the dendrogram at which the number of proteins in a cluster plateaus (Fig. 5B). Such plateaus identify levels of the global scFPs dendrogram at which single-cell features are practically inseparable, a division threshold that we call a root cluster (see Methods). For example, our adaptive thresholding method identified a root cluster corresponding to the nuclear pore complex, which distinguished it from other nuclear periphery proteins (Fig. 5B).

We compared the adaptive thresholding method to other clustering approaches from three different families—connectivity (hierarchical clustering (Murtagh and Contreras, 2012)), centroid (k-means (Sculley, 2010)) and density methods (DBSCAN (Ester et al, 1996)) (Fig. 5C). For each of the methods used in our comparison, we tried a range of hyperparameters and selected the ones that maximized median F1-score (see Methods). Evaluation was performed on a set of 140 protein complexes that contain at least three proteins included in the ORF-GFP localization dataset (Meldal et al, 2015). Our approach outperformed other methods in terms of four different scores—fold enrichment, F1 score, precision, and recall (Fig. 5C). The distributions of scores highlight the advantages of our adaptive thresholding approach. Density-based clustering fails at the protein complex identification task due to the high density of the feature space. At the same time, k-means fails at the identification of larger protein complexes (more than 5 protein members), hence its violin plot has two peaks. Hierarchical clustering is a more advantageous approach for this task, yet it requires information on the distance threshold and lacks adaptability for the protein complex size and cellular compartment. In contrast, our adaptive thresholding approach finds an optimal distance threshold for each cluster and, hence, it can discover protein complexes of varying sizes.

Using the adaptive thresholding clustering approach, we constructed a list of 88 high-confidence clusters whose proteins were indistinguishable at the single cell level (Dataset EV4; Fig. 5D). We mapped each cluster to a protein complex with the maximal fold enrichment and saw a median fold enrichment of 36.5 across 88 clusters, which is a 3-fold improvement over our clustering of aFPs with an optimized cut-off (Fig. 2). Of the 88 predicted clusters, 42 captured members of 32 different protein complexes distributed across 15 subcellular compartments (Dataset EV4). The remaining clusters did not capture two or more members of the same protein complex, although in 25/45 cases they contained proteins with PPIs (as annotated in BioGrid (Stark et al, 2006)). By using proteins from the same localization as our background set to compute fold enrichment, we tested whether the clusters could differentiate a protein complex from others in the same subcellular location (see Methods). We discovered that PIFiA confidently predicted members of protein complexes in multiple compartments, such as: [1] the proteasome and Ada2/Gcn5/Ada3 transcription activation complex in the nucleus; [2] LSM2-7 complex, decapping complex, and translation initiation factor 2B complex in the cytoplasm; [3] the oligosaccharyl transferase and Sec62-Sec63 complexes in the ER; [4] vacuolar proton translocating ATPase complex, phosphatidylinositol 3-kinase complex and iron exporter complex in the vacuolar membrane; [5] F-actin capping protein complex and PAN1 actin cytoskeleton-regulatory complex in the actin cytoskeleton; [6] Spc105 complex and NDC80 complex in the spindle pole; [7] retromer complex and SNX4-SNX41 sorting nexin complex in endosomes (see Dataset EV4).

In some cases (17/44), we identified all members of a complex, together with some additional proteins, which may be previously unappreciated complex components or members of an extended functional module, such as regulators or target proteins. For example, cluster #6 contained all 4 subunits of COMA, a kinetochore complex that connects proteins bound to centromeric DNA with those bound to microtubules, as well as nine additional proteins, eight of which display protein–protein interactions (PPIs) with COMA members (Stark et al, 2006). Other clusters identified proteins with the same biological role that may participate in PPIs. For example, cluster #39 contained 26 proteins that localized to the nuclear periphery in a punctate fashion. This group contained the only two GFP-tagged members of the TREX-2 complex, which couples SAGA-dependent gene expression and transcription elongation to mRNA export at the nuclear pore complex. Cluster #39 also included 8 proteins reported to have PPIs with members of the TREX complex (Stark et al, 2006), suggesting they may function in concert with the complex. Among the remaining proteins were members of a silencing complex, including Sir2, Sir3, Sir4 and the Sir4-interacting protein Esc1, which suppress transcription at subtelomeric regions, tethering them to the nuclear periphery (Deshpande et al, 2019). Thus cluster #39 identified proteins with roles in gene expression that localize to the nuclear periphery, some of which function to modulate each other's activity.

To provide some context into the limitations of the method, we looked at what types of protein complexes were under-represented in our set of high-confidence predictions. Identification by PIFiA was independent of protein abundance or number of members in the complex. The protein complexes we identified were enriched for localizations in small organelles and compartments (peroxisome, actin, nuclear periphery, endosome, spindle pole) and depleted for those in large diffuse compartments such as nucleus and cytoplasm. We note that not all protein complexes will have a distinct localization.

Using PIFiA features we have identified protein complexes from most compartments in the cell.

In summary, we have developed a novel approach for discovery of functional modules using solely the self-supervised feature profiles and leveraging the properties of microscopy data for optimal clustering, and prediction of molecular interactions.

## Interpretation of PIFiA features

Our analysis shows that PIFiA feature profiles contain condensed information about protein function at various levels of granularity. However, since deep neural networks function as 'black-box' models, it is difficult to dissect feature profiles and explain how individual features are related to the input images. To attempt to interpret PIFiA features, we first quantified feature importance for 15 different localization categories covering a diverse set of subcellular morphological patterns. We used the LR model described earlier to derive importance scores for each feature; the coefficients of the trained LR quantify how much each feature is predictive of a certain localization (Fig. 6A). Most localizations had more than three strongly predictive features (LR coefficient value > 5), suggesting that PIFiA learns to detect several distinctive patterns for each subcellular compartment. This confirms that PIFiA learns localization patterns with its convolutional filters, despite being trained on a completely different self-supervised objective. Also, the same feature could be predictive for several localizations (for example, features #3 and #28 recognize circular patterns corresponding to the vacuolar membrane and nuclear

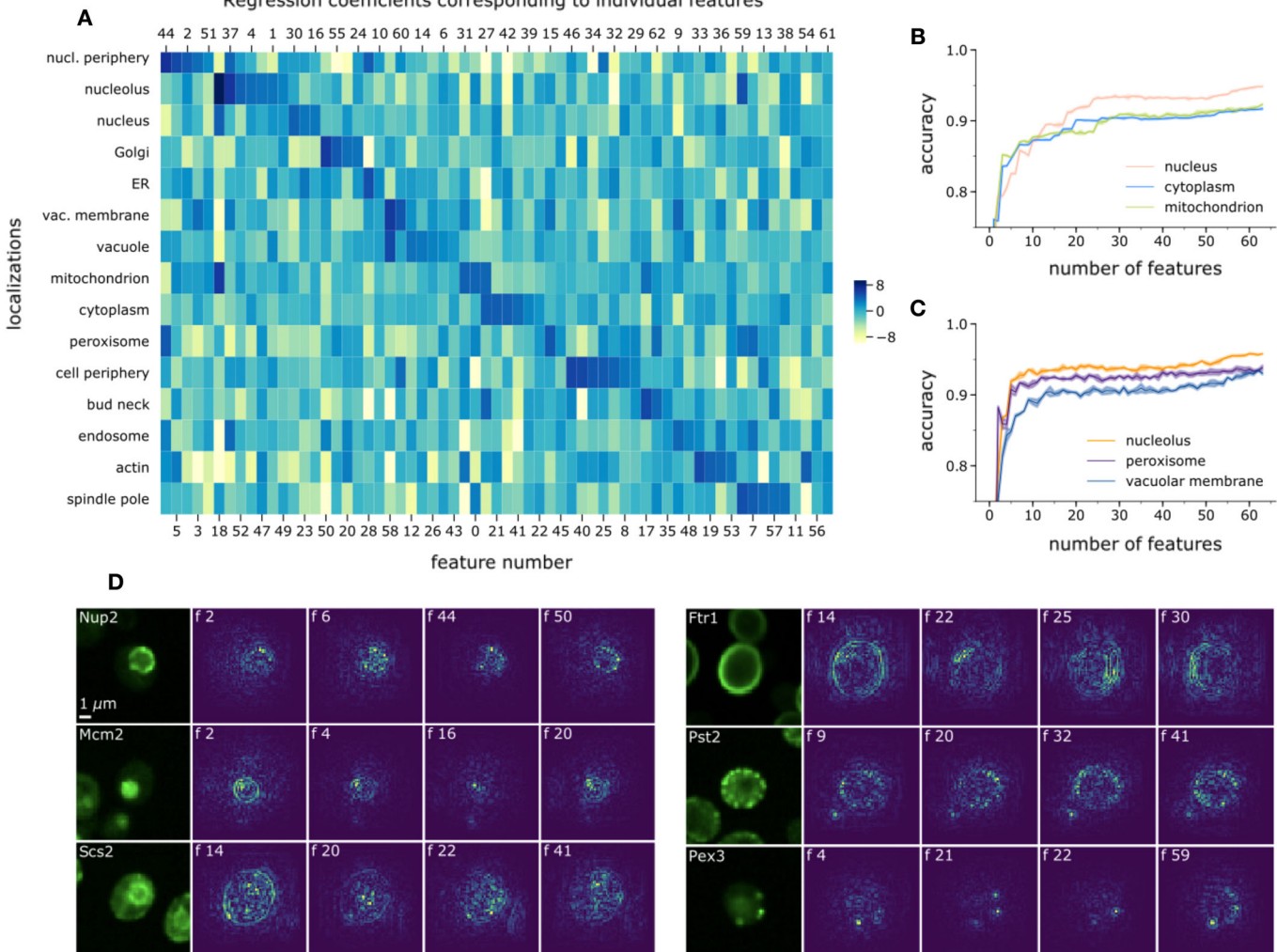

**Figure 6. Interpretation of PIFiA features (see also Appendix Fig. S3).**

(A) Heatmap of the logistic regression coefficients for localization prediction associated with each feature. Each feature was mapped to a localization (based on the maximal coefficient value), and features were sorted in descending order of coefficients. (B) Plot of classification accuracy dependence on number of features used to train the logistic regression for the three largest and most complex localizations: nucleus, cytoplasm and mitochondria. (C) Plot of classification accuracy dependence on number of features, showing three smaller and more homogeneous localizations: nucleolus, peroxisome and vacuolar membrane. Shading shows standard deviation across 5 runs. (D) Gradient maps highlighting regions of the input image that CNN "pays attention to". Proteins from six distinct subcellular compartments are shown with their most visually distinct gradient maps with the highest activation values.

periphery), or react to some variation of visual patterns present in multiple localizations. Overall, larger and more complex compartments required more features to be confidently classified. To illustrate this finding, we plotted classification accuracy for the three largest compartments (cytoplasm, nucleus, and mitochondria), as well as three homogeneous compartments (nucleolus, peroxisome, and vacuolar membrane) with respect to the number of features used during LR training (see Methods) (Fig. 6B,C; Appendix Fig S3A). While larger localization categories required approximately 30 features to reach their best performance, smaller localizations reached a saturation point at around 10 features.

Another way to interpret features learned by a CNN is to find regions of the image that had a large influence on the final result (Selvaraju et al, 2016; Zeiler and Fergus, 2014). Using gradient calculations, importance scores can be assigned to the input image pixels depending on the degree to which they affect the classification result or individual feature values (see Methods). We used the SmoothGrad approach (Smilkov et al, 2017; Zeiler and Fergus, 2014) to construct gradient maps for several features of the same protein image. We selected proteins representing five distinct subcellular localizations: Nup2 from nuclear periphery, Mcm2 from nucleus, Scs2 from ER, Ftr1, and Pst2 from cell periphery, and Pex3 from peroxisome. For each of the proteins, we used its scFPs to calculate 64 gradient maps for each of the individual features and selected four visually distinct gradient maps with the highest activation values for illustration purposes (Fig. 6D; Appendix Fig. S3B).

We observed that different features of the same image resulted in different gradient maps. While regions with higher intensity often have higher gradient values (unless they are not meaningful for the training objective), the interesting observation here is which part of these regions are meaningful for the particular feature. For example, the last gradient map of Pex3-GFP protein shows punctate localization patterns with three distinct punctae. Interestingly, different features react to different parts of the image—feature 4 reacts to the lower dot on the image and feature 23 reacts to two other dots. Also, gradient maps of the Mcm2 protein highlighted the nuclear periphery region, focus points in the nucleus and nuclear background signal. Similarly, different features of Ftr1 reacted to various subregions of the cell periphery. Of note, the generated gradient maps showed that the region of network attention was always the single central cell of the crop even for crops containing more than one cell, confirming that per-crop feature profiles are in fact single-cell profiles (Fig. 6D, with Mcm2, Ftr1 and Pst2 proteins containing multiple cells in their crop, Appendix Fig. S3B). Gradient-based interpretability approaches are useful to explain the relationship between individual features in the feature profile vector and input pixels in the image, and they constitute an important component of our downstream analysis pipeline. Hence, despite PIFiA's self-supervised training objective, we can visually understand what each learned feature represents in terms of the input image regions.

## Discussion

We describe PIFiA, a self-supervised computational workflow that learns protein functional signatures from single-cell fluorescence microscopy data. Feature profiles learned by PIFiA show state-of-

the-art performance on a variety of biological functional benchmarks, outperforming existing approaches for protein representation learning. Notably, our approach does not require any labels or annotations during training and uses only a single fluorescent channel. Hence, PIFiA can be easily applied to virtually any imaging dataset. We pre-trained PIFiA on a large-scale dataset encompassing over three million single-cell images of yeast cells expressing 4049 GFP-fusion proteins—a scale comparable to that of the commonly used computer vision dataset, ImageNet (Deng et al, 2009). As with ImageNet, we show that the yeast ORF-GFP dataset is a source for high-quality representation learning, enabling PIFiA to learn universal feature profiles that can be used out-of-the-box or minimally fine-tuned to suit an external standard. Thus, PIFiA can accelerate the rate of supervised training on external tasks by producing feature profiles that can fit any downstream task with simple linear regression, replacing multiple task-specific convolutional networks.

The PIFiA workflow unites a self-supervised convolutional neural network with multiple techniques for downstream feature profile analysis. The key advantage of our self-supervised objective is its independence of human annotations and its ability to learn high-quality features and ignore imaging artifacts and cell positions (Kobayashi et al, 2022; Razdaibiedina et al, 2019; Sullivan et al, 2018). To ensure that PIFiA remembers solely the biologically relevant patterns, yet ignores cell positioning and replicate noise, we require the network to learn the actual GFP-tagged protein by predicting its identity. Overall, we show that features learned by PIFiA outperform another self-supervised method, Paired Cell Inpainting (Lu et al, 2019), that was used to analyze the yeast GFP collection and even reaches the performance of the supervised approach, DeepLoc (Kraus et al, 2017), in its target task.

We describe downstream analysis techniques that use PIFiA feature profiles to explore different levels of subcellular organization that span both protein-level and single-cell feature profiles. Of note, our analysis focuses not only on the construction of a whole-proteome hierarchical map, but also provides quantitative rules to obtain clusters corresponding to a specific level of cellular organization, such as subcellular localization or biological process, and to identify proteins with multiple localizations and interacting proteins. This type of unbiased analysis can reveal unexpected properties and potential functions of proteins that can be further explored experimentally. For example, we used PIFiA features taken from images of yeast cells expressing GFP-tagged proteins to identify sub-compartmental groups enriched for proteins with biological processes not previously known to have distinctive subcellular localization patterns (Fig. 3E,F). We found that in addition to the known pan-nuclear localization, proteasome components are also localized at the nuclear periphery, a result we confirmed with co-localization experiments (Fig. EV5). Nuclear periphery localization of proteasomes has not been reported in yeast, but an in situ cryo-electron tomography study in *Chlamydomonas* found nuclear 26S proteasomes crowding around nuclear pore complexes (Albert et al, 2017). The role of proteasomes at the nuclear periphery may be to regulate transcription and/or to degrade proteins transiting the nuclear pore complex (Albert et al, 2017).

We also identified a group of proteins that localized specifically at the cell periphery of mother cells and were depleted from the growing bud. Budding yeast divide asymmetrically, with a replicative lifespan of

20–30 generations, where each division gives rise to a daughter whose replicative lifespan is reset and a mother who continues to age (He et al, 2018). Mother-specific cell periphery localization is achieved when three conditions are fulfilled (Eldakak et al, 2010). First, mother-specific proteins lack diffusive mobility in the plasma membrane. Second, newly synthesized proteins are deposited specifically in the growing bud. Third, the genes encoding these mother-specific proteins are expressed late in the cell cycle, so for cells in S/$G_2$ (small-budded cells) protein is detectable only in the mother. These steps ensure that the new and old pools of these proteins become spatially segregated during asymmetric division. Indeed, mother-specific localization of cell periphery proteins has been proposed to play a role in aging, with the daughter cell getting the newly synthesized copies of the protein, and the older and potentially more damaged copies inherited by the aging mother (Eldakak et al, 2010). Our set of asymmetrically segregating proteins includes 7 proteins previously seen to have mother-specific localization, plus 14 novel mother-specific proteins, including other transporters, proteins with roles in signaling, and 3 uncharacterized cell surface proteins (Dataset EV2). It is possible that accumulation of old and damaged versions of these newly identified proteins may also play a role in mother-specific aging.

We also applied PIFiA features for the identification of interacting proteins and members of protein complexes. To accomplish this, we used an adaptive thresholding method for single-cell clustering that exploits the biological properties of protein–protein interactions and microscopy data, outperforming conventional clustering methods for identifying members of protein complexes. We show that proteins whose single-cell PIFiA features are indistinguishable can be members of the same protein complex, have PPIs with each other, or have functionally related biological roles. A similar approach, *cytoself* (Kobayashi et al, 2022), was used to visually separate protein complexes from different compartments in human cells. Like the PIFiA pipeline, its self-supervised training scheme requires no pre-existing knowledge or categories, allowing it to reveal a highly resolved protein subcellular localization atlas that summarizes the major scales of cell organization (Kobayashi et al, 2022). Both the *cytoself* study and our work validate image-based feature profiles for downstream studies of protein organization and function in eukaryotic cells, both yeast (this work) and human cells (Kobayashi et al, 2022). Notably, we demonstrate that PIFiA can distinguish protein complexes from the same compartment in yeast cells, which are 5 to 30-fold smaller in size than human cells, providing a quantitative approach for downstream analysis and identification of functionally related proteins. Another study has implemented a similar representation learning paradigm to learn feature profiles of the images of cells with chemical or genetic perturbations (Moshkov et al, 2024) where perturbations were used as training labels. Thus, perturbation phenotypes were learned as inner representations from the network, suggesting that such a training scheme is effective for various types of experiments and biological systems.

In summary, the PiFiA pipeline extracts high-quality functional information about proteins from cell images in a quantitative form, without relying on pre-existing labels or manual annotations. In essence, the approach performs in silico colocalization, when two or more biological entities, such as proteins, are analyzed for similarity based on their respective localization patterns or positions within a cell (Dunn et al, 2011). In contrast to experimental colocalization, which is time-consuming and

expensive, in silico colocalization can be performed within seconds for multiple proteins at a time. In the case of PIFiA, this can be achieved not only with high speed but also with remarkable precision. Overall, PIFiA can be used to identify properties of proteins in single cells, including similarity and variability, that have the potential to inspire new experiments to uncover novel biological insights.

# Methods

## Construction of mutant arrays for imaging

For imaging screens, BY5299 (*MATα his3Δ1 leu2Δ0 ura3Δ0 met15Δ0 lyp1pr::TDH3pr-E2-Crimson::HPH::lyp1Δ can1pr::TDH3pr-tdTomato-NLS::URA3::can1Δ::STE2pr-LEU2*) was used as the starting query strain. E2-Crimson and td-Tomato-NLS are used as cytosolic and nuclear markers, respectively. The starting strain was crossed to the *MATa* ORF-GFP (Huh et al, 2003) and haploid strains carrying both the red fluorescent protein markers and the ORF-GFP were selected using the SGA method (Tong and Boone, 2006). All SGA selection steps were conducted at 30 °C, except sporulation, which was conducted at 22 °C for 10 days. The screen was performed in two biological replicates. We successfully constructed strains with 4049 (97.4%) GFP-tagged genes out of 4156 strains in the ORF-GFP collection. The missing GFP strains include those in linkage groups for *CAN1* and *LYP1*, as occurs in all SGA-derived collections. Other missing strains were those we were unable to grow from our original stocks. The missing strains had no bias for protein abundance (Ho et al, 2018). By GO cellular component, they were enriched for mitochondrial *cytochrome complex* (6 genes; Bonferroni corrected *P* value = 0.000368).

## High-throughput microscopy

Yeast cultures were prepared for microscopy and imaged as previously described (Chong et al, 2015; Cox et al, 2016; Mattiazzi Usaj et al, 2020). Briefly, haploid wild-type *MATa* strains expressing fluorescent protein fusions from SGA final selection plates were grown at 30 °C in low fluorescence synthetic minimal medium with Geneticin (200 μg/mL) and Noursoethricin (100 μg/ml). Cells were transferred to 384-well PerkinElmer CellCarrier Ultra imaging plates and centrifuged for 1 min at 500 g before imaging. Micrographs were obtained on an Opera Phenix (PerkinElmer) automated spinning disc confocal microscope. All imaging was done with a 63× water immersion objective. GFP was excited using a 488 nm laser and emission collected through a 520/35 nm filter. tdTomato was excited using a 561 nm laser, and emission collected through a 600/40 nm filter. E2Crimson was excited using a 640 nm laser, and emission collected through a 690/50 nm filter.

## Image acquisition for co-localization experiments

Protein pairs were chosen for co-localization if they had similar abundance (Ho et al, 2018) and localized to the same general subcellular compartment. For each protein, C-terminal fusions to both mNeonGreen and mScarlet were constructed as previously described (Meurer et al, 2018). Haploid cells in both configurations were mated to construct a/α diploids containing proteins tagged with the two fluorescent proteins. Diploid cells were grown and

imaged in low fluorescence synthetic minimal media (Sheff and Thorn, 2004) supplemented with Hygromycin B (300 mg/mL), Geneticin (200 mg/mL), and 2% glucose. Cells were grown at 30 °C to mid-logarithmic phase and transferred to Concanavalin A-coated 384-well PerkinElmer CellCarrier Ultra imaging plates. Images were acquired at 22 °C using the Opera Phenix (PerkinElmer) automated spinning disc confocal microscope. Three image fields of 5 Z-stacks of optical sections 0.7 μm apart were taken for each well. Each field contained 100–150 cells, acquired using the 63× water immersion objective. mNeonGreen was excited using the 488 nM laser, with emissions collected through a 520/35 nm filter. mScarlet-I was excited using the 561 nm laser, with emissions collected through a 600/40 nm filter. Digital Phase Contrast was used for cell detection using LED bright field imaging. All images were assessed by visual inspection.

## Dataset overview and image preprocessing

Images of the 4049 strains expressing a GFP-tagged protein visible above background fluorescence were obtained using an automated confocal microscope as described above. Cell images were obtained from two biological replicates, each of which had four fields of view for each GFP-tagged strain.

As the first step of preprocessing, we computed cell centers' coordinates across all images in the dataset using the nuclear channel. We obtained coordinates of the cells' centers by segmenting the nuclear channel with a simple Watershed algorithm and computing $x$, $y$ coordinates of the center of each cell's nucleus (McQuin et al, 2018). We ignored cells with centers too close to the crop's boundary (less than 10 pixels). Based on the cell center coordinates, we created single-cell crops of 64 × 64 pixels around those centers across all images in the dataset. We filtered crops that had GFP signal intensity less than the 5th percentile of the whole-proteome GFP intensity distribution, and crops dominated by the background noise (i.e., a uniform signal across the whole crop, with variance). After filtering low-quality crops, we dropped proteins with less than 10 cells, and we obtained 3,058,961 unique cells in the dataset. Then, the dataset was split into training, validation, and test sets using 80%, 10%, and 10% of the cells of each protein, respectively. The training subset contained 2,450,801 single-cell crops, and validation and test subsets contained 304,080 single-cell crops each. Finally, we applied instance normalization by standardizing the raw pixel intensities of every crop to a mean of 0 and a variance of 1 (independently for each channel of each sample). PIFiA was trained on 64 × 64 pixel crops of the GFP channel. During training, we used random flipping (horizontal and vertical) and random rotation across (0, 90, 180, 270) degrees to augment the training data. Labels of the training set are one-hot class vectors of length 4049.

## Architecture and training

The architecture details are illustrated in Fig. S1A. The backbone of PIFiA consists of eight convolutional blocks followed by three fully-connected layers. Each convolutional block consists of a convolutional layer, batch normalization and rectified linear unit activation. Training was performed using Adam optimizer (Kingma and Ba, 2014) with a learning rate of 1e−3 and cosine decay learning rate schedule (number of steps equal to the number of training updates

during 30 epochs), with cross-entropy as an objective function ($y_i$ and $\hat{y}_i$ are predicted probability and ground truth label of the protein $i$; $N$ is the total number of classes, i.e.,4049 proteins):

$$L(\hat{y}_i, y_i) = -\sum_{i=1}^{N} y_i \log \hat{y}_i$$

To prevent overfitting, we applied dropout regularization (Srivastava et al, 2014) of 0.05 (5% dropout rate) after the second fully-connected layer (feature extraction layer). We performed hyperparameter optimization and selected the learning rate from {1e−4, 3e−4, 5e−4, 8e−4, 1e−3, 3e−3, 5e−3} and dropout rate from {0.01, 0.02, 0.05, 0.1, 0.2, 0.3, 0.5} based on maximal validation accuracy. Network parameters were initialized using a truncated normal distribution function with a standard deviation of 0.1. To report the performance, we ran the model three times with different random weights initializations; each run was 30 epochs and model weights were saved after every epoch. All the experiments were performed in Python using Tensorflow. The model was trained on the computing cluster of the Vector Institute for Artificial Intelligence, using NVIDIA T4 GPU with 12GB of VRAM, and up to 32GB of system RAM (single CPU). Source code and usage examples are available at https://github.com/arazd/pifia.

We used early stopping to select the final model (Girosi et al, 1995). We defined stopping criteria based on the model's test accuracy of proteins classification across 4049 protein classes. We stop at an epoch where a derivative of the test accuracy becomes smaller than a threshold of 0.5% for at least 3 epochs, i.e., a point at which accuracy starts to saturate (Fig. S1B). Our goal was to stop at the point when the model has already grasped the most important morphological patterns, yet highly related and interacting proteins are not distinguished from each other. This trend is further illustrated by plots of average precision, F-score and precision (we show 0.9 threshold) for protein complexes and pathways standards (Fig. S1B). With protein prediction accuracy increasing over the course of training, the precision improved, but after some epochs, AP and F-score either saturated or started to decline. We found that accuracy saturation thresholds between 0.2% and 0.7% yielded comparable and optimal solutions, though other stopping points can be used depending on the training schedule, as well as model applications and goals. The proposed early stopping strategy helped to prevent memorizing noise and unnecessary patterns, while retaining morphologically similar proteins close in the feature space.

## Benchmarking and baseline feature extraction

We compared performance of feature profiles learned by PIFiA to features from three other popular methods for protein representation learning/extraction—DeepLoc (Kraus et al, 2017), Paired Cell Inpainting (Lu et al, 2019), and CellProfiler (McQuin et al, 2018).

A classic modular feature extraction tool, Cell Profiler (McQuin et al, 2018), was applied to the GFP and cytoplasmic channels of the test images across 4049 GFP-tagged proteins. We obtained 433 pre-defined CellProfiler features that quantitatively measure cellular phenotypes, including intensity, shape, and texture. Since some of the CellProfiler features can be repetitive, its representations are often post-processed with Principal Component Analysis (PCA) (Abdi and Williams, 2010). In our work, we evaluated both the

original CellProfiler representation with 433 individual features, and its PCA projection (37 individual features) that explains 99% of the variance.

We used the DeepLoc model by Kraus et al, (2017) as our supervised learning baseline. We investigated two training modes: (a) with per-protein localization labels from Huh et al, (2003); (b) with single-cell localization labels manually annotated in our lab. Per-protein annotations can sometimes be subjective and might not capture the nuances of protein localization at a single-cell level accurately. Such annotations apply to all of single-cell images of a protein and are available for a significant part of the yeast proteome from the Huh et al, study (Huh et al, 2003). In contrast, single-cell labels were manually annotated for the individual cell images by research scientists in our lab. While more precise, such labels are expensive and encompass a much smaller portion of the proteome. The original DeepLoc version uses single-cell labels. For a fair comparison, we used both single-cell and protein-level localization labels to train DeepLoc and reported the corresponding results.

Hence, we used two model variants: (a) the original version of DeepLoc (which was trained on a set of 21,882 single-cell crops with manually assigned labels; pre-trained weights provided by Kraus et al, (2017), and (b) our adaptation of DeepLoc, DeepLoc+PIFiA, which was re-trained on a larger set of 1,432,774 images with less accurate protein-level labels instead of expensive yet more precise single-cell labels.

We trained DeepLoc+PIFiA from scratch on the GFP channel of the same training set images using 1,432,774 single-cell crops from 15 one-hot localization categories derived from Huh et al, (2003). We performed hyperparameter optimization and selected the most optimal learning rate from {1e−4, 3e−4, 5e−4, 8e−4, 1e−3, 3e−3, 5e−3} and dropout rate from {0.01, 0.02, 0.05, 0.1, 0.2, 0.3, 0.5}. We chose 3e−4 learning rate with cosine decay learning rate schedule and 0.05 dropout rate based on maximal validation accuracy. The model was trained with Adam optimizer for 30 epochs (model weights were saved every epoch for subsequent evaluation), with cross-entropy as an objective function, $y_i$, $\hat{y}_i$ are predicted probability and ground truth label of localization $i$, total $N = 15$ localization classes):

$$L(\hat{y}_i, y_i) = -\sum_{i=1}^{N} y_i \log \hat{y}_i$$

We also used early stopping to select the final DeepLoc+PIFiA model weights. DeepLoc+PIFiA model selection was based on maximal validation set accuracy. Network parameters were initialized using a truncated normal distribution function with a standard deviation of 0.1. We performed 3 runs with different random weights initializations and performed training with a batch size of 128. After training, we extracted features of the test set images from the last hidden layer of the DeepLoc+PIFiA model following previous studies (Razdaibiedina and Brechalov, 2022).

For our self-supervised learning baseline, we used the Paired Cell Inpainting method (Lu et al, 2019). Contrary to other models, Paired Cell Inpainting requires two channels for training—cytoplasmic background and target protein; hence we performed training of Paired Cell Inpainting using the GFP and cytoplasmic channels of the test images across 4049 GFP-tagged proteins. We used the exact same architecture and training objective described by Lu et al, (2019). The objective function minimizes a standard pixel-

wise mean-squared error loss between the predicted target protein $\hat{s}_t$ and the actual target protein $s_t$ ($h$ and $w$ are pixels across image width and height, respectively):

$$L(\hat{s}_t, s_t) = \frac{1}{h \cdot w}(\hat{s}_{h,w,t} - s_{h,w,t})^2$$

We performed hyperparameter optimization and selected an optimal learning rate of 1e−4 from {1e−4, 3e−4, 5e−4, 8e−4, 1e−3, 3e−3, 5e−3}. The Model was trained with Adam optimizer for 30 epochs (3 runs in total), and model weights were saved every epoch for subsequent evaluation. We selected the final model with early stopping based on the minimal validation set loss. After training, we extracted feature profiles of the test set images by maximum pooling the output of an intermediate convolutional layer, across spatial dimensions, as suggested by Lu et al, (2019).

## Evaluation of aFPs

Functional benchmarks used for assessment of the quality of the resulting feature profiles were derived from Gene Ontology Cellular Component (Harris et al, 2004) (4045 protein annotations), Gene Ontology Slim Biological Process (Harris et al, 2004) (3968 protein annotations), KEGG pathways (Kanehisa and Goto, 2000) (1422 protein annotations) and EMBL protein complexes (Meldal et al, 2015) (1402 protein annotations). Dubious ORFs and proteins without annotations were left out during comparison. To evaluate resulting features without further fine-tuning, we used strategies from two distinct perspectives: information retrieval and clustering quality.

Following standard practice, we computed pairwise distances across all available aFPs (4049 × 4049 distances in total) and sorted them from highest to lowest. Then protein pairs (which were not left out) were marked as positive if they had the same annotations, or negative otherwise, and AP and F-scores were computed. For proteins to be considered a positive pair, we required an exact agreement between labels in case of pathways and protein complexes standards, while for GO annotations we required at least 50% of the labels to overlap (due to high quantity of assigned labels). Results reported in Fig. 2B,C are based on ranking aFP pairs with correlation distance, and we found similar trends when using euclidean and cosine distances. We chose to continue analyses with the correlation metric due its lower susceptibility to fluctuations in individual feature values, and hence higher tolerance to outliers, which is a desirable property for the PIFiA workflow. For the clustering-driven benchmark, we clustered aFPs and compared clusters to the sets of proteins annotated to a certain term, and for each standard (we required cluster size to be at least 2 to be informative). For comparison, we applied AMI score (Vinh et al, 2010) between the resulting clusters and protein groups related to a certain term (with a higher score indicating more agreement between clusters and standard-defined categories). To obtain an AMI score for each method, we performed hierarchical clustering (with average linkage and correlation as a distance) of its per-protein representations and derived clusters across all similarity thresholds between 0.1 and 0.95 with a step of 0.05, and reported the maximal AMI across clusterings. For each deep learning model, feature profile evaluation was performed across 3 runs (results shown with bar plots in Fig. 2).

## Visualization of PIFiA feature profiles (aFPs and scFPs)

We used t-SNE (Van der Maaten and Hinton, 2008) for visualization of PIFiA feature profiles. We set the perplexity parameter to 40 for visualization of whole-proteome feature profiles averaged on the per-protein level (~4000 points) (Fig. 3C,D,F), and to 200 for visualizing single-cell feature profiles from the test set (>100,000 points) (Fig. 5A). We represented the distribution of fundamental GO bioprocesses with a kernel-density estimate (KDE) using Gaussian kernels (Fig. 3D). We applied outlier filtering by removing points that do not lie within two standard deviations from the mean (across $x$ or $y$ t-SNE coordinates). We used Scott's rule for KDE bandwidth selection (Scott, 1979).

## Hierarchical clustering with aFPs

We performed agglomerative hierarchical clustering (Murtagh and Contreras, 2012) of the whole-proteome aFPs (4049 in total) using correlation as a distance metric and average linkage. The optimal cut-off distance for the whole-proteome hierarchical clustering was determined using the AMI curve between clustering labels and provided standard annotations, following the diminishing returns principle to find the elbow point. At the optimal distance cutoff, the slope of the curve becomes negligible, indicating that the available clusters cover most of the standard's functional groups. In Fig. 3 we used GO Cellular Component annotations (Harris et al, 2004) as a standard and calculated clustering labels at different correlation thresholds between 0 and 1, with a step of 0.01. Clustering was performed on the whole-proteome feature profiles, and AMI scores were calculated on a subset of proteins that had a single annotation according to GO Cellular Component. We identified an optimal cut-off point when the derivative of the AMI curve (calculated over 20 steps, starting at correlation of 1) was less than a threshold of 0.1. The proposed strategy can be used on different standards, without requiring annotations to cover all proteins (Appendix Fig. S1A).

## Training logistic regression

To perform localization mapping, we trained a multinomial logistic regression (LR) using single-cell feature profiles obtained with PIFiA from the training set. We used supervised labels from 17 manually annotated localizations defined by Huh et al, (Huh et al, 2003) (we left out "ambiguous" category and classes with 5 or less proteins), and limited our training set to proteins that had a single annotated localization. Overall, our training set consisted of 1,432,774 single-cell feature profiles and included 2415 proteins from mitochondrion (465), nucleus (472), cytoplasm (799), actin (27), ER (245), vacuole (95), bud neck (8), spindle pole (35), Golgi (15), peroxisome (20), vacuolar membrane (47), cell periphery (51), nuclear periphery (45), endosome (28), and nucleolus (63). We followed our previously described dataset split (each protein's single-cell crops were split into train, validation, and test sets with 8:1:1 ratios). We used NVIDIA T4 GPU with 12GB of VRAM, and up to 16GB of system RAM on a single CPU to accelerate training; we trained LR for 5 epochs using Adam optimizer (Kingma and Ba, 2014) (1e−3 learning rate) and cross-entropy as a training loss; LR weights were saved after every epoch and we selected the final LR model with early stopping based on maximal validation set accuracy. Of note, after 2 epochs LR predictions stabilized and the difference between subsequent models was minimal (less than 3% test accuracy deviations).

To evaluate the quality of LR predictions, we compared its test set performance with DeepLoc (Kraus et al, 2017) (training procedure described in the benchmarking section). DeepLoc and LR were trained and evaluated on the sets of the same size and protein composition, with the only difference being DeepLoc used image crops for training, while LR using self-supervised scFPs from the pre-trained PIFiA model. Precision-recall curves for PIFiA LR and DeepLoc were generated on unseen scFPs and corresponding images from the test set of the single-localizing proteins (Fig. EV2B).

## Sub-compartmental clustering with aFPs

We performed sub-compartmental clustering using single-localizing aFPs from the test set that were classified to the same localization by the previously described LR. For an aFP to be single-localizing, we required that its highest softmax probability was at least 0.6, and second-highest was no greater than 0.2 (more detailed analysis of single-localizing proteins and localization heterogeneity is described in the next section). We clustered aFPs of proteins that mapped to the same localization category and produced 15 per-compartment hierarchical trees (we used average linkage and correlation distance for clustering).

We calculated the Silhouette score (Rousseeuw, 1987) using scikit-learn library, as the mean intra-cluster distance ($a$) and the mean nearest-cluster distance ($b$) for each aFP. The Silhouette coefficient for a sample is $(b-a)/\max(a, b)$; $b$ is the distance between a sample and the nearest cluster that the sample is not a part of. We surveyed 15 per-localization hierarchical trees, clustered with average linkage and correlation metric, using correlation thresholds between 0.25 and 0.75, with a step of 0.05. Median of Silhouette scores across all localizations for a given threshold is shown in Appendix Fig. S1E. We found that thresholds between 0.5 and 0.6 yield maximal Silhouette scores, and a distance threshold of 0.5 corresponded to maximal GO bioprocess AMI value for whole-proteome aFPs clustering. Hence, we chose a 0.5 threshold to cluster aFPs belonging to each per-localization tree, and obtained 30 clusters, which we subsequently called sub-compartmental groups.

## Analysis of localization heterogeneity with scFPs

We used scFPs from the test set to analyze whole-proteome localization heterogeneity patterns. First, we used the pre-trained LR on 17 localization categories (described in the previous section) to map each protein's test set scFPs to one of the 17 localization classes. We observed that the most probable localization class had an average 0.74 probability per protein (computed across all test set scFPs), while 2nd and 3rd classes scored 0.11 and 0.052 per-protein probabilities, respectively. This motivated us to perform heterogeneity analysis with the two most probable localization categories, to avoid low scFP quantities and potential noise effects. We then computed a mean probability across each localization class to determine the two most frequent localizations of the protein. Hence, for each protein X we obtained a distribution of 2-dimensional real-valued probability vectors $[p_{i1}, p_{i2}]$ with $p_{i1}$ and $p_{i2}$ corresponding to the probabilities of the first and second

most frequent localization classes, $i \in \{1, \ldots, N\}$, $N$ is the number of test set scFPs of the protein X. Given this distribution, we could compute whether protein X is single-localizing or has AND-type/OR-type localization heterogeneity. We filtered low-confidence scFPs $i$, whose sum of probabilities was below a confidence threshold: $p_{i1} + p_{i2} < \alpha_{conf}$ (low-confidence region). Next, based on a heterogeneity threshold $\beta$, we divided the rest of the scFPs into first localization if $p_{i1} > p_{i2} + \beta$, second localization if $p_{i2} > p_{i1} + \beta$, or mixed-localizing category otherwise. We varied values of $\alpha_{conf}$ between 0.5 and 0.9, and values of $\beta$ between 0.25 and 0.75 (with a step of 0.05), inspecting numbers of assignments into localization categories and low-confidence region, and selected $\alpha_{conf}$ and $\beta$ as 0.5 based on elbow point analysis. Hence, scFPs of each protein were mapped into one of four classes—primary localization, secondary localization, mixed localization or low-confidence region (Fig. 4D). If we assume that percentages of the corresponding categories for protein X are $c_1$, $c_2$, m, k (class 1, class 2, mixed and low-confidence, respectively), then protein X would be marked as AND-type localizing if the mixed category had a higher percentage of scFPs than primary and secondary localizations together: $m > c_1 + c_2$; otherwise protein X would be marked as OR-type if no less than 8% of scFPs belonged to the secondary localization: $min(1, c_2) > 0.08$, and single-localizing in the other case. We experimented with OR-type thresholds between 0.05 and 0.3 (with a step of 0.01), and found that the number of OR-type localizing proteins saturated between 0.07 and 0.1 thresholds. We selected 0.08 as an elbow point between $min(1, c_2)$ value and number of category assignments. Thus, each protein was marked as single-localizing, OR-type, AND-type, or undetermined (if too many scFPs were assigned as low-confidence).

## Cell cycle prediction and annotation with scFPs

We trained an ensemble of three CNNs for cell cycle classification using cytosolic and nuclear channels from our dataset. These two channels contained enough information to distinguish the cell cycle stage of a cell. The CNN contains four convolutional blocks followed by two fully-connected layers, and was trained to predict one of four cell cycle stages - G1, S, metaphase and anaphase (MA), or telophase (T) (Appendix Fig. S2A).

We manually labeled 800 crops of cells from 103 different proteins, corresponding to distinct cell cycle stages (with 200 crops from each class) according to bud emergence; we used heavy data augmentation during training to prevent overfitting: rotation by arbitrary angle, vertical and horizontal flips, image zoom within 0.02 range, and vertical and horizontal shifts of up to 9 pixels. We used three-fold cross-validation. The training was performed on $64 \times 64 \times 2$-dimensional crops over 150 epochs using Adam optimizer, with loss being a categorical cross-entropy across 4 cell cycle categories. We performed hyperparameter optimization to select learning rate (5e−4) and dropout rate (0.05). We performed 3 independent runs with random weights initialization (using truncated normal distribution with a standard deviation of 0.1). Model weights were saved after every epoch, and final models for each run were selected with early stopping based on maximal validation accuracy. Training and test accuracy and categorical cross-entropy loss are shown in Appendix Fig S1. We created an ensemble of three CNNs (from epochs corresponding to minimal loss value), and subsequently mapped each single-cell crop to a 4-dimensional real-valued vector of cell cycle probabilities. The cell cycle probability vector was computed as an average of the probability vectors of three CNNs of that crop. We subsequently joined T and G1 categories due to high cell density in certain crops, which could potentially lead to an incorrect cell cycle category assignment.

We applied Mann–Whitney U test (McKnight and Najab, 2010) to identify proteins whose localization changes had cell cycle dependency. For each protein with localization heterogeneity, we annotated its single-cell crops from train, validation and test sets using both LR localization categories and cell cycle stages (via cell cycle classifier). We selected two primary annotated localizations, and compared cell cycle stage distribution of the corresponding crops. For each cell cycle stage, our null hypothesis was that the stage was equally represented among both localizations. Localizations with significant distribution differences (i.e. $p$-value $< 1e-3$) were annotated as related to the particular cell cycle stage.

## Functional enrichment analysis

Gene Ontology (GO) enrichments were performed using GO-term Finder Version 0.86, available through the *Saccharomyces cerevisiae* Genome Database (https://www.yeastgenome.org/goTermFinder). We applied gene set enrichment analysis (GSEA) using Python package GSEApy (Kuleshov et al, 2016) (https://github.com/zqfang/GSEApy) to analyze hierarchically clustered protein groups, sub-compartmental groups and sets of multi-localizing and mixed-localizing proteins (Dataset EV2 and EV3). Query gene sets for GSEA included GO biological process, cellular component, and molecular function standards (Harris et al, 2004). GSEA results were filtered to include gene sets with $p$-values below 0.05 and a minimum gene set size of 2. We applied Bonferroni correction to obtain adjusted $p$-values. We also applied one-sided Fisher's exact test with Costanzo group 19 (Costanzo et al, 2016) categories to analyse nucleus OR cytoplasm, nucleus AND cytoplasm gene sets, reporting protein sets with $p$-value $< 0.05$ as the ones showing enrichment (Dataset EV3).

## Discovery of interacting proteins from scFPs

We identified clusters containing potentially interacting proteins using two steps. First, we hierarchically clustered scFPs from the test set (we used average linkage and a correlation metric). After that, we divided the dendrogram from top to bottom and traced the number of unique proteins inside the cluster along with the division thresholds of 0.05 points. We found thresholds of the dendrogram at which the number of proteins in a cluster plateaus (95% of protein composition remains the same). After such "morphologically inseparable" clusters were identified, we used three data-driven scores to measure the quality of the resulting clusters—cell ratio, elbow point, and child ratio. Cell ratio $c$ is an average percentage of a protein's cells that fall into a particular cluster. A higher cell ratio translates into less dispersed cells of the same protein, and more confident protein assignment into the particular cluster. Elbow point $k$ is a clustering distance at the level of the current cut (1-PCC in our case). A lower elbow point corresponds to a smaller distance between proteins in their feature profiles space. Descendant ratio $d$ of the particular root cluster is the percentage of its descendent clusters that were annotated to the same root. A high child ratio corresponds to more agreement of the

child clusters, hence indicating a more confident prediction. We devise a final score $s$ as follows:

$$s = \frac{c \cdot d}{k}$$

We used a final score cutoff of 0.6 to produce a list of 88 high-confidence clusters. We compared performance of our adaptive thresholding approach with clustering approaches from three different families—connectivity (hierarchical clustering (Murtagh and Contreras, 2012)) centroid (k-means (Sculley, 2010)) and density methods (DBSCAN (Ester et al, 1996)) (Fig. 5C). We performed clustering on the same test set of scFPs with different approaches. For each of the methods used in our comparison, we tried a range of hyperparameters (k ranging from 5 to 500 with a step of 5 in k-means, epsilon ranging from 0.1 to 5 with a step of 0.1 in DBSCAN, and correlation threshold ranging from 0.05 to 0.5 with a step of 0.025 for hierarchical clustering) and report the ones corresponding to the maximal median F1-score across all clusters. F1 scores were calculated by assigning each pair of scFPs ground truth label (0 or 1 depending on whether they are part of the same protein complex) and predicted label (0 or 1 depending on whether they are part of the same cluster).

### Gradient maps

We applied the SmoothGrad method to obtain per-feature gradient maps of the input images (Smilkov et al, 2017). Original gradient maps $m_c(x)$ compute the derivative of activation function $S$ of the highest-scoring class $c$ with respect to the input image $x$, and thus highlight pixels which influence classification decision:

$$m_c = \frac{\partial S_c(x)}{\partial x}$$

Since we were interested in feature interpretation, but not interpreting the classification decision, we modified this computation. In our implementation of gradient map $M_i(x)$, we take the derivative of specific feature $f_i$ from the feature vector $f$ with respect to the input image $x$:

$$M_i(x) = \frac{\partial f_i(x)}{\partial x}$$

Hence, our gradient maps highlight regions of the image that impact the value of the selected feature. SmoothGrad produces a gradient map $M_i(x)$ by averaging a number of gradient maps obtained from an input image with added noise from Gaussian distribution $N(0, \sigma^2)$ (with a mean 0 and a standard deviation $\sigma$):

$$\hat{M}_i(x) = \frac{1}{n} \sum_{i=1}^{n} M_i(x + N(0, \sigma^2))$$

We used $n = 100$ images and $\sigma = 0.05$ noise level.

### Generalization experiments

We performed generalization experiments by applying PIFiA out-of-the-box on two unseen yeast imaging datasets: CYCLoPS (Koh et al, 2015) and YeastRGB (Dubreuil et al, 2019).

CYCLoPS is a collection comprising more than 20 million cells of C terminal-tagged GFP images of 4144 proteins. To derive single-cell crops from this dataset, we applied the Watershed algorithm to the cytoplasmic channel and computed $x$, $y$ coordinates of the center of each cell (McQuin et al, 2018) subsequently making $64 \times 64$ pixel crops around the centers. We only used GFP channel of the single-cell crops and performed per-image standardization of each crop.

The YeastRGB dataset is a collection of GFP-tagged microscopy screens produced using SWAT technology, where new-generation fluorescent reporters are fused at the N' and C' of open reading frames of over 4000 proteins. We used C'-tagged images (and excluded N'-tagged images) from the YeastRGB dataset to avoid performance mismatch related to the tag location since our training data has C'-tagged images. The YeastRGB database provides single-cell crops, from which we only used GFP channel images and performed per-image standardization of each crop.

For each dataset, we run single-cell crops through PIFiA and extracted scFPs. Next, we averaged scFPs to obtain aFPs for all proteins in the dataset. We used t-SNE (Van der Maaten and Hinton, 2008) for visualization of aFPs with perplexity = 40 and color-coded the resulting maps with the Huh et al, localization standard (Huh et al, 2003).

## Data availability

The image data used in this work are available at the CellVision website (https://thecellvision.org/pifia/): The raw images: https://thecellvision.org/pifia_files/pifia_raw_data.tar.gz. The dataset of single-cell cropped images: (https://thecellvision.org/pifia_files/pifia_single_cell_crops.tar.gz). Source code for the PIFiA network and downstream analysis is available at https://github.com/arazd/pifia.

## Peer review information

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

## Acknowledgements

We thank Oren Kraus, Michael Costanzo, Nil Sahin, Alan Moses, Leah Cowen, and Matej Usaj for valuable discussions and advice. This work was supported by grants from the National Institutes of Health (R01HG005853 to BA and CB), and the Canadian Institutes of Health Research (PJT-180259 to BA). Equipment for automated image acquisition and analysis was purchased using funds from the Canadian Foundation for Innovation and the Ontario Research Fund. JB was supported by the Canadian Institute for Advanced Research (CIFAR) AI Chairs program and the National Sciences and Engineering Research Council (Canada). We gratefully acknowledge the support of NVIDIA Corporation with the donation of the Titan Quadro P6000 GPU. Computational resources were provided, in part, by the Province of Ontario and the Government of Canada through the Vector Institute for Artificial Intelligence. AR was supported by the Province of Ontario (Ontario Graduate Scholarship, 2021–2022) and the Vector Institute for Artificial Intelligence (Vector Institute Postgraduate Affiliate Scholarship, 2019–2021). CB is a Fellow of the CIFAR.

## Author contributions

**Anastasia Razdaibiedina**: Conceptualization; Software; Formal analysis; Validation; Investigation; Visualization; Methodology; Writing—original draft; Writing—review and editing. **Alexander Brechalov**: Conceptualization; Software; Supervision; Methodology; Writing—original draft. **Helena Friesen**: Conceptualization; Formal analysis; Supervision; Validation; Methodology; Writing—original draft; Project administration; Writing—review and editing. **Mojca Mattiazzi Usaj**: Supervision; Writing—review and editing. **Myra Paz David Masinas**: Software. **Harsha Garadi Suresh**: Resources. **Kyle Wang**: Resources. **Charles Boone**: Supervision. **Jimmy Ba**: Conceptualization; Supervision; Investigation; Methodology. **Brenda Andrews**: Supervision; Funding acquisition; Writing—review and editing.

## Disclosure and competing interests statement

The authors declare no competing interests. Brenda J Andrews is a member of the Advisory Editorial Board of Molecular Systems Biology. This has no bearing on the editorial consideration of this article for publication.

# Expanded View Figures

**Figure EV1.  PIFiA network architecture and training settings.**

**Related to Fig.** 1. (**A**) Overview of the architecture of PIFiA convolutional network. (**B**) Left plot: test accuracies of three different runs over the course of training (X axis: epochs, Y axis: test accuracy). Smaller plots: average precision, F-score and precision on protein complexes and pathways standards (X axis: epochs, Y axis: corresponding score on test set). The purple line indicates point of early stopping, when accuracy starts to saturate (derivative of the test accuracy smaller than a threshold of 0.5%). (**C**) Bar graphs comparing the current PIFiA architecture with a common baseline, DenseNet-121, across four different standards (Gene Ontology Cellular Component, Gene Ontology Bioprocess Slim, KEGG Pathways, EBI Protein complexes) in terms of average precision, F-score and adjusted mutual information (assessed on aFPs of 4049 proteins). Error bars represent standard deviation from the mean across three network runs. (**D**) Bar graphs comparing PIFiA performance across different dimensions of the feature profiles (32, 52, 64, 80, 128).

▶

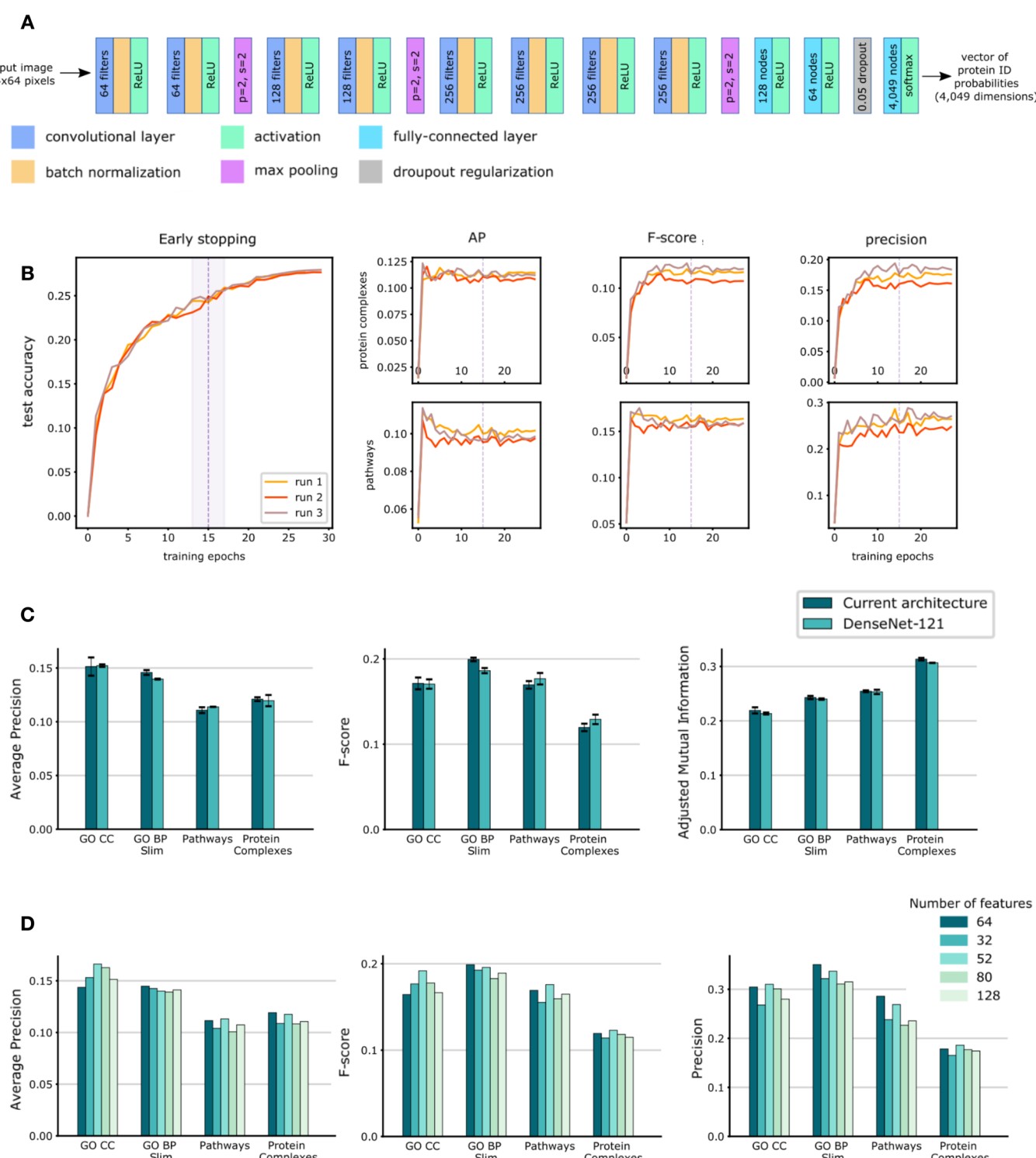

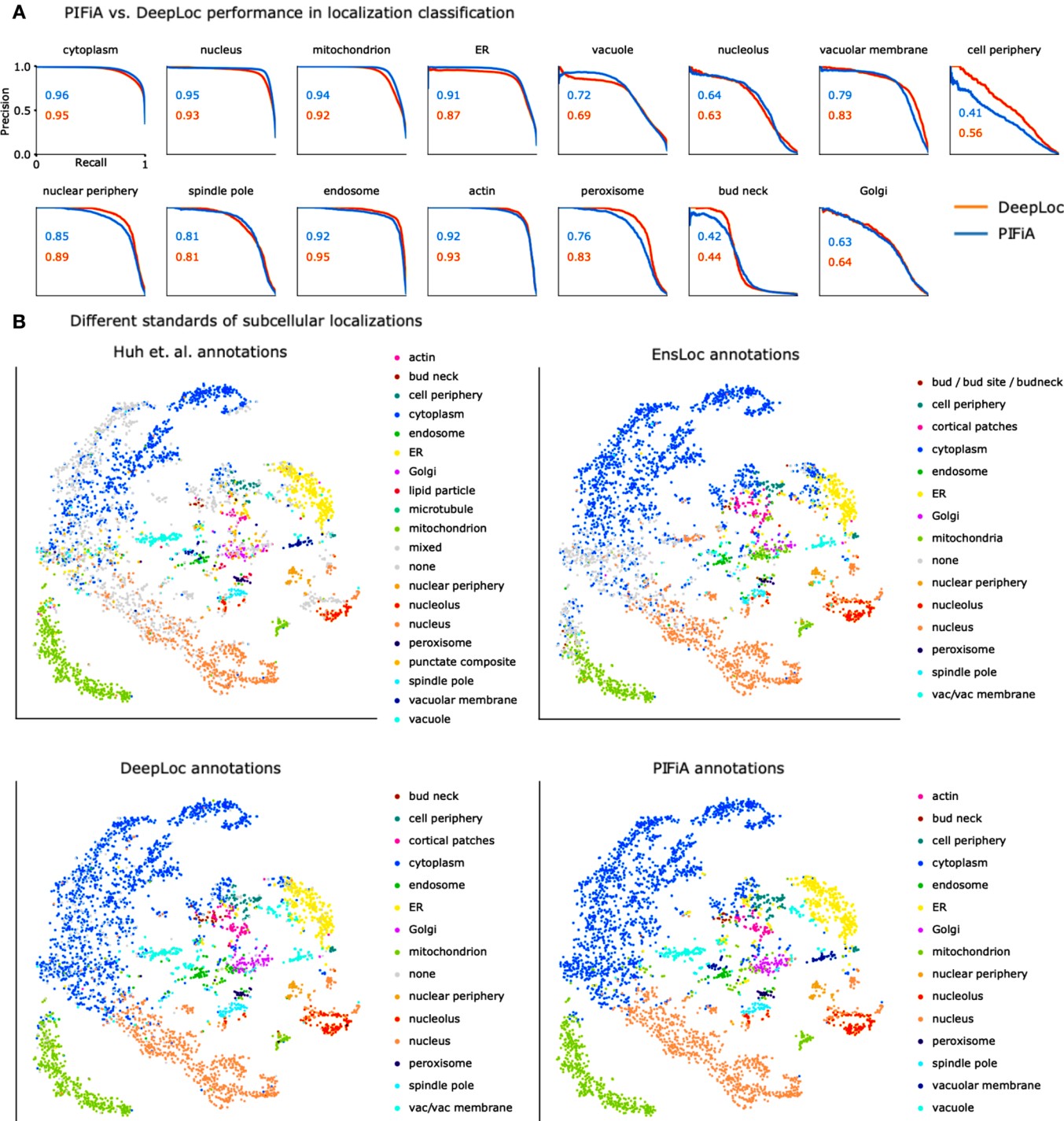

**Figure EV2. Comparison of PIFiA annotations and existing localization standards.**

Related to Fig. 3. (A) Comparison of localization classification performance of DeepLoc versus PIFiA feature profiles coupled with a logistic regression. Precision-recall plots are shown for 15 subcellular localizations. (B) Whole-proteome aFPs tSNE colored by different annotations of subcellular localization: manual annotations from Huh et al, (2003) (Huh et al, 2003), and computationally-derived annotations from EnsLoc, DeepLoc and PIFiA.

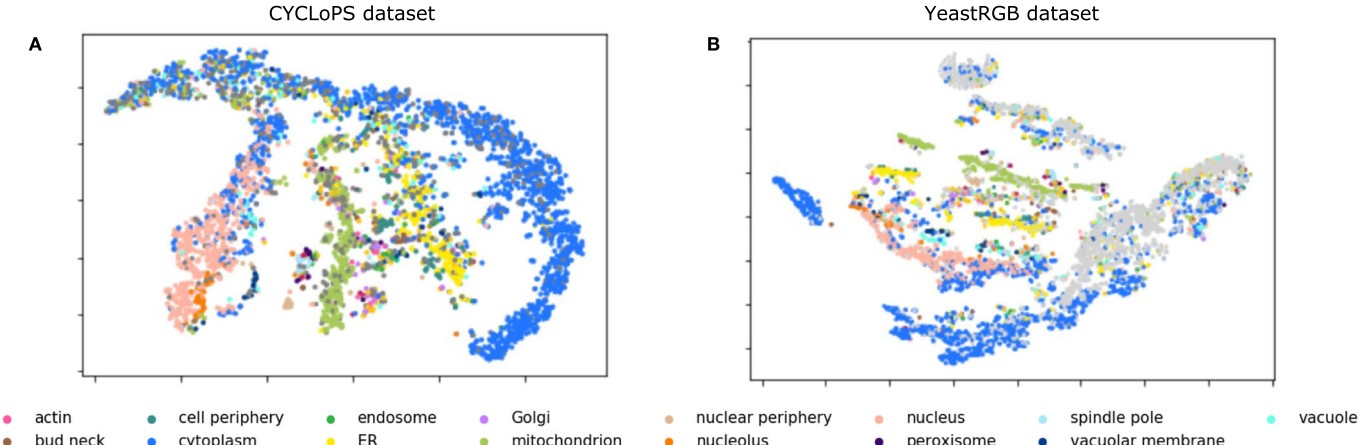

**Figure EV3.   Generalization of PIFiA network to two unseen datasets.**

**Related to Fig. 3.** **(A)** Localization-colored tSNE on aFPs obtained from the CYCLoPS dataset. **(B)** Localization-colored tSNE on aFPs obtained from the YeastRGB dataset.

Examples of subcompartment groups

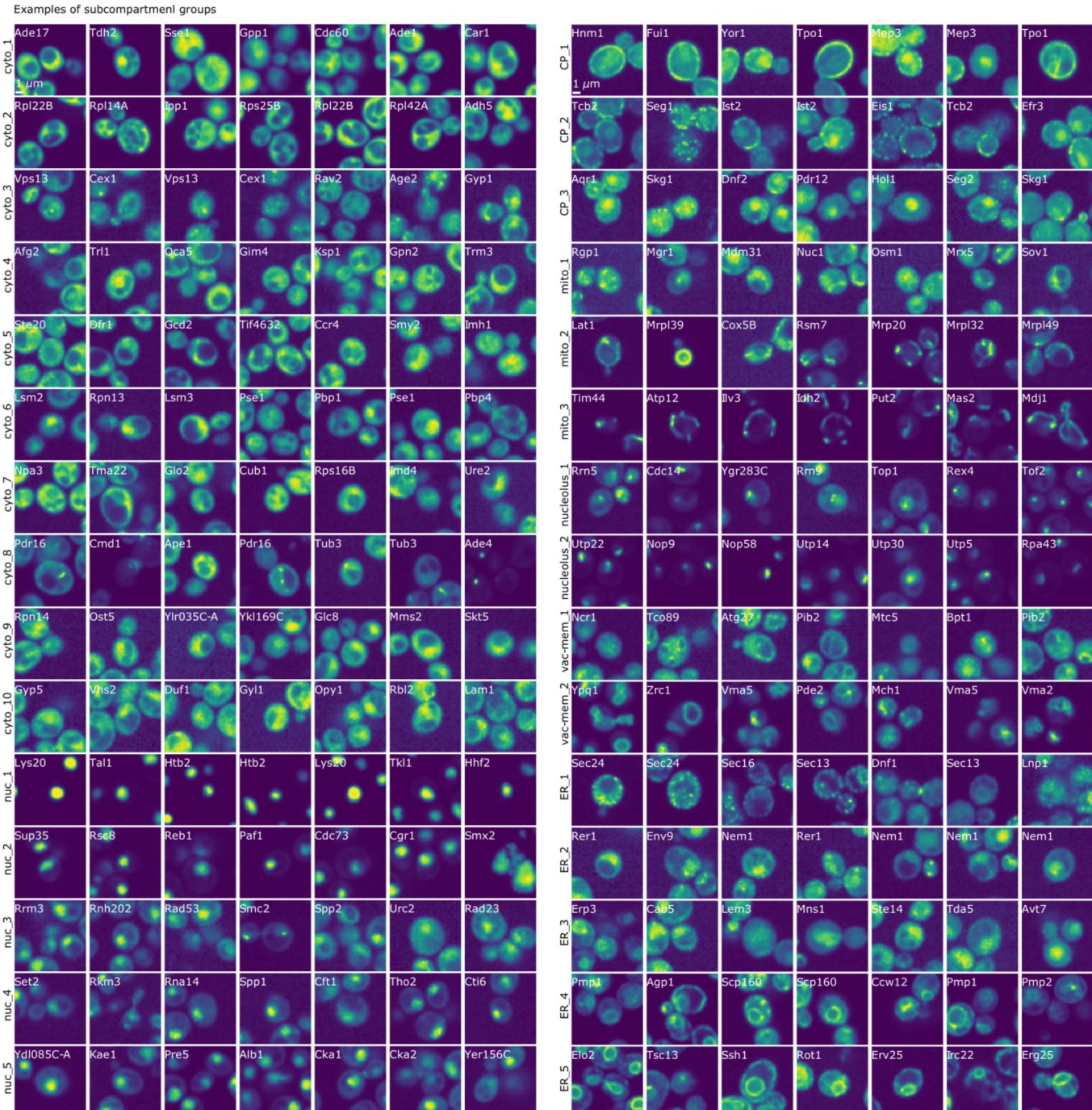

**Figure EV4. Examples of proteins from 30 different sub-compartmental groups.**

**Related to Fig. 3.** Each row corresponds to a sub-compartmental cluster (e.g. nuc-1, nuc-2). The relevant GFP-tagged protein is identified on each micrograph.

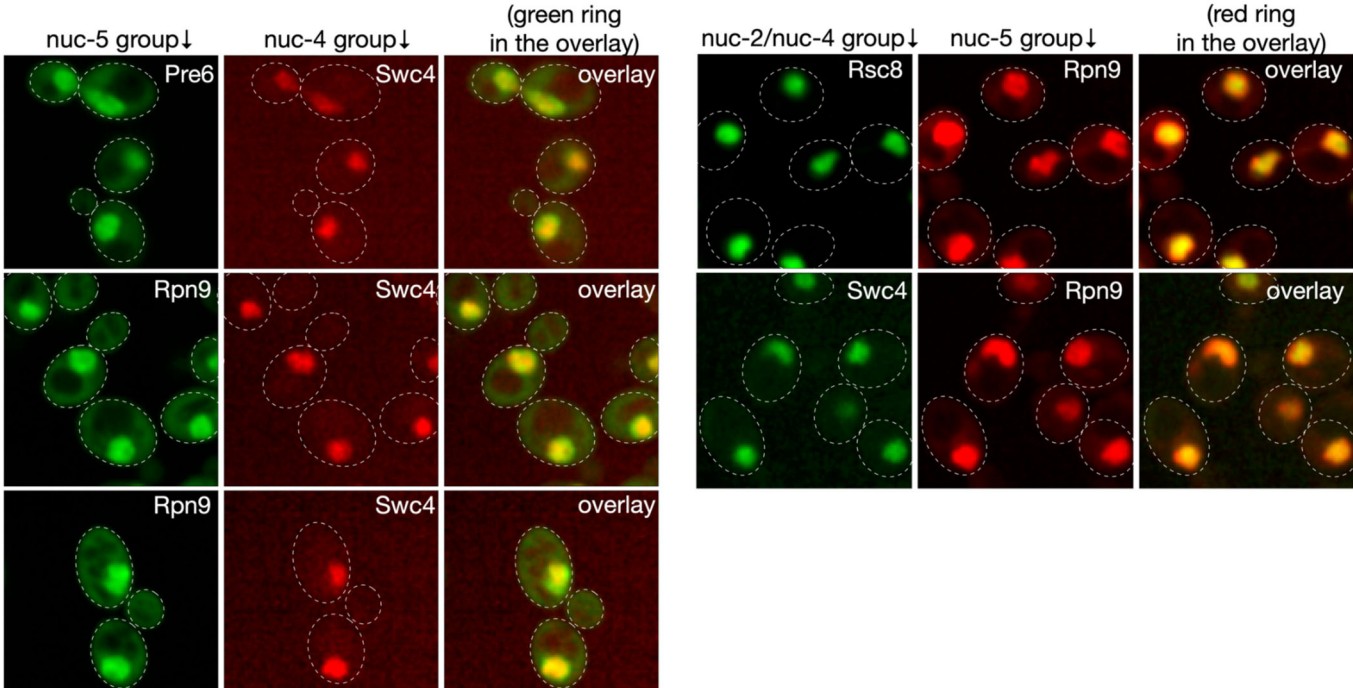

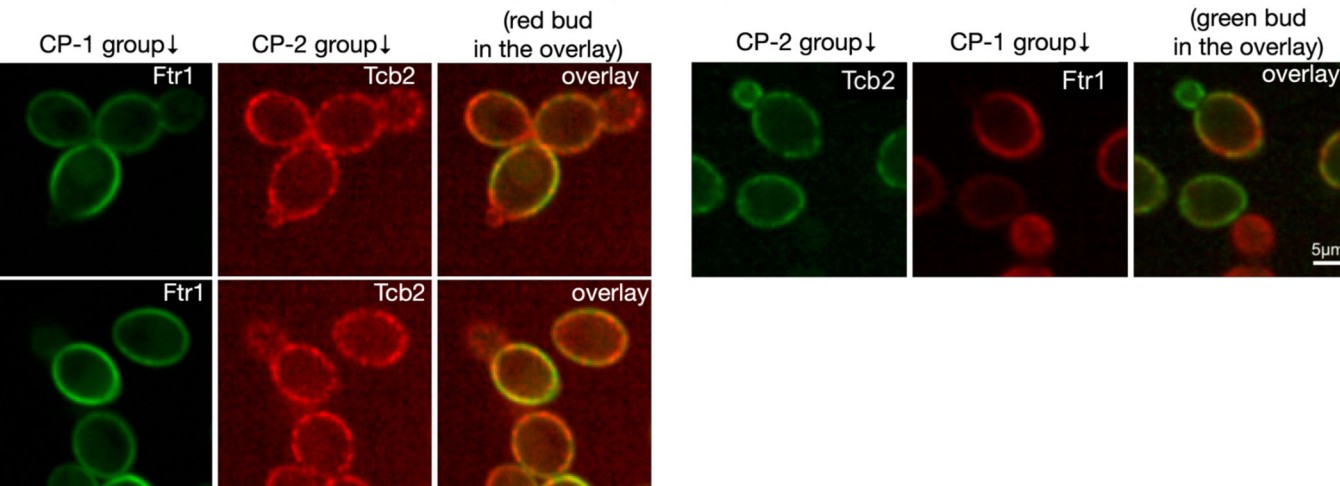

**Figure EV5. Colocalization assay for proteins from different sub-compartmental.**

Related to Fig. 3. Colocalization experiment results: representative micrographs of cells expressing mNeonGreen- (green images) or mScarlet- (red images) tagged proteins annotated to nucleus (top panel) or cell periphery (bottom panel) groups. Overlays of the mNeonGreen and mScarlet images are shown on the right of each triplet of images. The tagged proteins are indicated on the micrographs (scale bar shown bottom right).

