## [Peer Review File · Molecular Systems Biology]

PIFiA: Self-supervised Approach for Protein Functional Annotation from Single-Cell Imaging Data

Brenda J. Andrews, Anastasia Razdaibiedina, Alexander Brechalov, Helena Friesen, Mojca Mattiazzi Usaj, Myra Paz Masinas, Harsha Garadi Suresh, Kyle Wang, Charles Boone, and Jimmy Ba

Corresponding author(s): Brenda J. Andrews (brenda.andrews@utoronto.ca) , Charles Boone (charlie.boone@utoronto.ca), Jimmy Ba (jba@cs.toronto.edu)

Review Timeline:

Submission Date:	15th Aug 23
Editorial Decision:	25th Sep 23
Revision Received:	24th Dec 23
Editorial Decision:	14th Feb 24
Revision Received:	27th Feb 24
Accepted:	28th Feb 24

Editor: Maria Polychronidou

Transaction Report:

25th Sep 2023

Manuscript Number: MSB-2023-11946

Title: PIFiA: Self-supervised Approach for Protein Functional Annotation from Single-Cell Imaging Data

Dear Brenda,

Thank you again for submitting your work to Molecular Systems Biology. We have now heard back from two of the three reviewers who agreed to evaluate your study. Unfortunately, after several reminders we have not received a report from reviewer #1. In the interest of time, we have decided to proceed with making a decision based on the two available reports. As you will see below the reviewers acknowledge that the study seems like a relevant contribution to the field. They do however raise a series of concerns which we would ask you to address in a major revision. Please note that reviewer #2 have also provided their report as a PDF (attached below), as it contains a figure.

Without repeating all the points listed below, one of the more fundamental issues raised by both reviewers refers to the need to strengthen and extend the biological insights derived by the performed analyses. Reviewer #2 mentions that this is particularly important given the somewhat limited advance from a methodological point of view. Both reviewers provide constructive suggestions on how to address this issue.

All issues raised by the reviewers need to be satisfactorily addressed. As you may already know, our editorial policy allows in principle a single round of major revision, so it is essential to provide responses to the reviewers' comments that are as complete as possible. Please feel free to contact me in case you would like to discuss in further detail any of the issues raised or if you would like to share your revision plan with me. I would be happy to schedule a call.

On a more editorial level, we would ask you to address the following points:

- Please include 5 keywords.
- Please provide a .doc version of the manuscript text (including legends for the main figures and EV Figures) and individual production quality figure files for the main Figures and EV Figures (one file per figure).
- We have replaced Supplementary Information by the Expanded View (EV format). In this case, all additional figures can be included in a PDF called Appendix. Appendix figures should be labeled and called out as: "Appendix Figure S1, Appendix Figure S2... Appendix Table S1..." etc. Each legend should be below the corresponding Figure/Table in the Appendix. Please include a Table of Contents in the beginning of the Appendix. For detailed instructions regarding expanded view please refer to our Author Guidelines: .
- Please provide a "standfirst text" summarizing the study in one or two sentences (approximately 250 characters), three to four "bullet points" highlighting the main findings and a "synopsis image" (550px width and max 400px height, jpeg format) to highlight the paper on our homepage.
- All Materials and Methods need to be described in the main text. We would encourage you to use 'Structured Methods', our new Materials and Methods format. According to this format, the Materials and Methods section should include a Reagents and Tools Table (listing key reagents, experimental models, software and relevant equipment and including their sources and relevant identifiers) followed by a Methods and Protocols section in which we encourage the authors to describe their methods using a step-by-step protocol format with bullet points, to facilitate the adoption of the methodologies across labs. More information on how to adhere to this format as well as downloadable templates (.doc or .xls) for the Reagents and Tools Table can be found in our author guidelines: . An example of a Method paper with Structured Methods can be found here:
- Please include a "Disclosure and Competing Interests Statement" in the main text.
- Please include a Data availability section describing how the data, code etc. have been made available. This section needs to be formatted according to the example below:
The datasets and computer code produced in this study are available in the following databases:
 - Chip-Seq data: Gene Expression Omnibus GSE46748 (<https://www.ncbi.nlm.nih.gov/geo/query/acc.cgi?acc=GSE46748>)
 - Modeling computer scripts: GitHub (<https://github.com/SysBioChalmers/GECKO/releases/tag/v1.0>)
 - [data type]: [full name of the resource] [accession number/identifier] ([doi or URL or identifiers.org/DATABASE:ACCESSION])
- For data quantification: please specify the name of the statistical test used to generate error bars and P values, the number (n) of independent experiments (specify technical or biological replicates) underlying each data point and the test used to calculate p-values in each figure legend. The figure legends should contain a basic description of n, P and the test applied. Graphs must

include a description of the bars and the error bars (s.d., s.e.m.).

- The References should be formatted according to the Molecular Systems Biology reference style.

When you resubmit your manuscript, please download our CHECKLIST (<https://bit.ly/EMBOPressAuthorChecklist>) and include the completed form in your submission.

Please note that the Author Checklist will be published alongside the paper as part of the transparent process (<https://www.embopress.org/page/journal/17444292/authorguide#transparentprocess>).

If you feel you can satisfactorily deal with these points and those listed by the referees, you may wish to submit a revised version of your manuscript. Please attach a covering letter giving details of the way in which you have handled each of the points raised by the referees. A revised manuscript will be once again subject to review and you probably understand that we can give you no guarantee at this stage that the eventual outcome will be favorable.

Kind regards,

Maria

Maria Polychronidou, PhD
Senior Editor
Molecular Systems Biology

We realize that it is difficult to revise to a specific deadline. In the interest of protecting the conceptual advance provided by the work, we recommend a revision within 3 months (24th Dec 2023). Please discuss the revision progress ahead of this time with the editor if you require more time to complete the revisions. Use the link below to submit your revision:

IMPORTANT: When you send your revision, we will require the following items:

1. the manuscript text in LaTeX, RTF or MS Word format
2. a letter with a detailed description of the changes made in response to the referees. Please specify clearly the exact places in the text (pages and paragraphs) where each change has been made in response to each specific comment given
3. three to four 'bullet points' highlighting the main findings of your study
4. a short 'blurb' text summarizing in two sentences the study (max. 250 characters)
5. a 'thumbnail image' (550px width and max 400px height, Illustrator, PowerPoint or jpeg format), which can be used as 'visual title' for the synopsis section of your paper.
6. Please include an author contributions statement after the Acknowledgements section (see <https://www.embopress.org/page/journal/17444292/authorguide>)
7. Please complete the CHECKLIST available at (<https://bit.ly/EMBOPressAuthorChecklist>).

Please note that the Author Checklist will be published alongside the paper as part of the transparent process (<https://www.embopress.org/page/journal/17444292/authorguide#transparentprocess>).

See also figure legend guidelines: <https://www.embopress.org/page/journal/17444292/authorguide#figureformat>

9. Please note that corresponding authors are required to supply an ORCID ID for their name upon submission of a revised manuscript (EMBO Press signed a joint statement to encourage ORCID adoption).

(<https://www.embopress.org/page/journal/17444292/authorguide#editorialprocess>)

Currently, our records indicate that the ORCID for your account is 0000-0001-6427-6493.

Link Not Available

The system will prompt you to fill in your funding and payment information. This will allow Wiley to send you a quote for the article processing charge (APC) in case of acceptance. This quote takes into account any reduction or fee waivers that you may be eligible for. Authors do not need to pay any fees before their manuscript is accepted and transferred to the publisher.

EMBO Press participates in many Publish and Read agreements that allow authors to publish Open Access with reduced/no

publication charges. Check your eligibility: <https://authorservices.wiley.com/author-resources/Journal-Authors/open-access/affiliation-policies-payments/index.html>

*** PLEASE NOTE *** As part of the EMBO Press transparent editorial process initiative (see our Editorial at <https://dx.doi.org/10.1038/msb.2010.72>), Molecular Systems Biology publishes online a Review Process File with each accepted manuscripts. This file will be published in conjunction with your paper and will include the anonymous referee reports, your point-by-point response and all pertinent correspondence relating to the manuscript. If you do NOT want this File to be published, please inform the editorial office at msb@embo.org within 14 days upon receipt of the present letter.

Reviewer #2:

See attached pdf (for figures etc.)

Razdaibiedina et al. present PIFiA, a weakly-supervised / self-supervised method that learns (without human annotations) representations of protein localization patterns, called "protein feature profiles". PIFiA relies on a pretext task of predicting protein identity directly from the fluorescently-labeled input image, without enforcing reconstructions of these images. The authors acquired a massive dataset of over 4,000 GFP-tagged fluorescently labeled proteins in yeast, applied PIFiA to extract quantitative protein profiles, and demonstrated applicability in (1) clustering by hierarchical representation by protein profile similarities, (2) predicting protein function based on profile similarity, (3) characterizing localization heterogeneity of a protein via PIFiA representations of single cells.

The manuscript is clearly written and relatively (to a technical paper) easy to follow. The manuscript is accompanied with open code (looks well organized and documented - I did not try to execute it) and the data is available via an interactive website for visualization and exploration.

The strengths of the manuscript are (1) Proper performance evaluations, which is not trivial in this domain of representation learning, using existing knowledge for functional benchmarks at different scales (GO, CC, Pathways, Protein Complexes), and co-localization analysis for validation, (2) using the weakly/self-supervised representations as inputs for supervised training, demonstrating that PIFiA representations can be used for downstream supervised tasks (where annotated data is limited) without training from scratch, (3) Characterizing protein localization heterogeneity using PIFiA's single cell representations, the analysis in Fig. 4 was new and creative and I can see it being replicated by others, (4) Resource, source code and interactive website.

Overall, I am mostly concerned regarding PIFiA's limited methodological novelty. The same concept, of self/weak supervision with the pretext task of predicting protein identity from the images, was presented in two earlier manuscripts here to encode perturbation outcomes and here for protein localization profiling (Cytoself). The Cytoself manuscript convincingly showed that enforcing image reconstructions improves the attained protein profile representations beyond that achieved with the pretext task which comes in contrast to the claims made here, but without sufficient evaluations to support these claims. More details follow below in the specific comments.

I do believe that the manuscript can be re-pivoted to focus on the resource and downstream analyses rather than on the method as the main contribution. This resource and analyses would enable validation of yeast protein localization/function, and generation of hypotheses regarding unknown protein functions and would make a nice contribution to the field.

Major comments and suggestions:

1. Methodological advancement and comparison to the state-of-the-art methods.

The method is highly similar to the one presented in <https://doi.org/10.1101/2022.08.12.503783> (which the authors should cite) that uses the same concept. Cytoself (<https://doi.org/10.1038/s41592-022-01541-z>) that was applied to a similar task of encoding protein localization profiles. The authors' main justification for their design choices was the argument that optimizing the pretext task, without enforcing reconstruction, creates better representations. This argument is made in page 3-4 ("A disadvantage of autoencoders is that they often learn features that are not relevant to protein morphology and localization, such as cell position, imaging artifacts and noise. Also, pixel-level reconstruction is computationally expensive and often unnecessary for representation learning. These issues reflect the autoencoder's training objective, which targets identical reconstruction of the input."). This argument was evaluated directly in the Cytoself paper, where it was shown that the reconstruction (decoder) contributes to effective representations (see Fig. 4 "Cytoself without decoder" copied below). The comparisons of PIFiA to other approaches (Cell Profiler, DeepLoc, Paired Cell Impainting, DeepLoc + PIFiA) are not sufficient to make this point, which should be evaluated explicitly by comparing the same model with and without reconstruction. If the authors want to make this argument, I recommend evaluating Cytoself with vs. without decoding (I am aware of other labs who were successful at using Cytoself's representations to analyze their data), or at least PIFiA, with vs. without decoding. If the decoder indeed deteriorates the representations then these conclusions, contradicting those of Cytoself, should be discussed.

Fig. 4 from the Cytoself paper:

2. Generalization across biological replicates. Is it necessary to learn new representations for each dataset or could learned representations effectively generalize across different datasets? This is an important point that is crucial for understanding the method's capacity and potential use by others, but is not evaluated in the manuscript. The dataset contains two biological replicates for each protein (with four fields of view per replicate). The authors could train PIFiA on one biological replicate and evaluate it on the other (and vice versa) to evaluate this point. Whether these representations can be transferred to other datasets should be discussed. It was demonstrated that Cytoself can perform well on an external independent dataset - this can be referenced in the discussion in this context.

3. The authors do not explicitly state whether the ORF-GFP yeast imaging dataset itself is one of the key contributions of this manuscript. Was the data collected for this manuscript? Would the raw and the processed data become publicly available? The visualization is a nice contribution, but the link to the files https://thecellvision.org/pifia_files is not working.

4. The text in the figures is tiny, which makes some of them hard to follow, especially Figures 3 and 5.

5. I was not able to fully follow the results reported in Figure 5. More details in the main text would be helpful.

Other comments and suggestions:

1. A more detailed explanation of the comparison to DeepLoc + PIFiA would be helpful. In page 19, the authors state that the benchmark was "re-trained on a larger set of images with less accurate protein-level labels instead of expensive yet more precise single-cell labels." - what is the meaning and implications of "less accurate labels"? Is it due to the human protein localization annotations?

2. Moreover, to ensure a fair comparison between the weakly/self supervised approach and supervised approaches, it is advisable to consider using the same PIFiA architecture for supervised training. While comparing DeepLoc+PIFiA as a benchmark is a valuable evaluation, it remains unclear whether the observed difference in results is derived from the difference in architectures or training. Thus, I suggest:

a. A more clear and comprehensive description on the training labels for the supervised approach. What were the limitations/noise in these labels?

b. Compare PIFiA trained with weak labels versus trained with protein-level labels, using the same architecture.

c. Assuming that the focus of the manuscript will not be the Method or that the authors will perform the Cytoself evaluation with/without the decoder, these evaluations are not necessary.

3. Fig. 2 legends, please provide explicit references to Kraus et al., and Huh et al.

4. Fig. 2. It would be helpful to plot the random prediction accuracy (horizontal line) as a baseline for context.

5. Fig. 3a - I recommend excluding the features visualization which does not provide any useful information.

6. Fig. 3d - the text and legend mentions KDE visualization, but contours are not present in the figure. In general, I would recommend revising Fig. 3 legend.

7. I found the images of the gradients in Fig. 6d not very convincing. What can be learned from regions with higher intensities that tend to have higher gradients, how does this support the claim made that "the CNN captures a variety of within-cell morphological patterns"

8. Comparison to Cytoself

a. Cytoself was demonstrated for downstream analysis and interpretation. This should be cited in this context (page #4 in the introduction). One example is Fig.

3b which is very similar to the spectral representations (Fig. 5) in Cytoself.

b. In the Discussion that authors say that "A similar approach, cytoself, was used to visually separate protein complexes from different compartments in human cells on tSNE.", downplaying the contribution of the Cytoself paper (which did more than just tsne visualization). Please report these contributions properly.

9. In the Discussion that authors refer to "in silico colocalization" a term that was never defined / explained.

10. Fig. S6 (cell cycle) is not referenced in the text.

Reviewer #3:

In this manuscript, Razdaibindina et al have developed PIFiA, an unsupervised classification method that converts input spinning disc confocal microscopy images of single cells into discrete categories that specify the subcellular compartments imaged proteins belong to and biochemical functions they participate in, similar to a previously developed unsupervised method (Paired Cell InPainting). PIFiA was applied to the genome-wide GFP library of budding yeast (a rebuilt library based on the Huh et al GFP library from Erin O'Shea's lab). At the heart of the method are 2 steps. First, the method employs a series of convolutional neural networks that associate pixel intensities in smaller subregions of the image in order to learn a reduced dimension representation of the features of the image (the "feature profile"). Second, the feature profile is input into a series of fully connected layers that lead to discrete classification of each protein based on its feature profile. The PIFiA average feature profiles classifications are then validated for relevance (e.g., by checking that highly correlated aFP's belong to the same subcellular compartment, etc.).

The most significant result of the manuscript is the apparent biological validity of the feature profiles extracted through unsupervised learning (i.e., that proteins can be clustered into functionally relevant categories based on image-based reduced

dimension representations extracted without outside knowledge like expert annotations). This association of proteins into functional classes spans multiple scales of organization, from organelles to membership in protein functional modules. The algorithm developed here is sound. The analysis is interesting, especially with regard to the suggestion of potential functionally relevant protein co-localization based on subtle, not pre-defined image features that would escape the capability of the human eye to uncover. Most of the classifications appear to be straightforward consequences of the compartmental nature of the eukaryotic cell and thus derived from the fact that most of the compartments are morphologically diverse organelles. As a tool to cluster proteins into distinct groups PIFiA appears to be useful and represents an advance in terms of using image data to its fullest extent compared to methods such as Cell Profiler that use ad hoc pre-defined features for dimensionality reduction. Biologically speaking, however, the most significant drawback, is the lack of validation of whether the patterns (especially the subtle patterns that are the highlight of this method) uncovered by PIFiA encode interactions of physiological importance or what the origin of these subtle localizations are.

Major questions:

1. One of the most surprising findings to me was the ability of PIFiA to distinguish diffraction-limited structures from each other given that structure was detected through CNN's. Do the authors have an explanation for why this method is effective in distinguishing peroxisomes or endosomes from COP vesicles for example (e.g., in Fig 5)? The authors went to some lengths to try and open the "black box" in Fig. 6 (an important exploration for which the authors are to be commended), but I think PIFiA would be even better tested by examining how subdiffraction structures are distinguished from each other. As a concrete suggestion (though the authors will undoubtedly come up with a better idea): what kind of feature profiles are learned upon masking of pixels contained within the brightest puncta when looking at peroxisomal (or Golgi or lipid droplet, etc) proteins? Do these feature profiles exhibit any plausible biological relevance (e.g., if one masks out the punctate signals from peroxisomal proteins, would gradient maps as performed in Fig. 6d have overlap with the ER, suggesting that PIFiA is able to pick up on subtle protein trafficking patterns to help it distinguish apparently diffraction limited mature structures?). In the case of peroxisomes, though, it appears not to be the case (e.g., in Fig. 4f there does not appear to be shared signal between peroxisomes and ER).

2. The other major potential of PIFiA as a discovery tool is its potential ability to use single cell feature profiles to assign proteins to functional modules based on subtle yet highly correlated feature profiles. These subtle features could represent substantial new insight into functionally relevant protein-protein interactions. But for this to be the case the authors should experimentally test how these subtle features change upon perturbations to the putative protein-protein interactions. The ideal type of perturbation would not be one that induces a gross mislocalization of a protein, but instead one that only weakens the physical association between two proteins in a complex and results in disruption to the subtle aspects of the feature profile correlations. As an example of my suggestion, Eugster et al EMBO J 19: 3905-3917 2000 (doi: 10.1093/emboj/19.15.3905) contains a dissection of COP I domains whose genetic ablation induces changes to COP architecture; the authors might know better examples where mutations disrupt binding surfaces between proteins that cluster together in the same functional module in PIFiA and could be employed to test their effect on subtle feature correlations.

Alternatively, the authors could provide an analysis of what fraction of proteins known to belong to functional modules but whose memberships in these modules are not successfully uncovered by PIFiA feature correlations to provide important context into the limitations of the method.

3. I imagine much of the measured success of PIFiA (in terms of aggregative metrics such as the F-score, AP, etc) lies in many successfully classified proteins belonging to morphologically distinct organelles (eg ER, mito, cytoplasm). To more finely gauge how much of an advance PIFiA is compared to the other methods tested here I would recommend reproducing Fig. 2 for cases where morphologies are not dramatically different. For example if the authors tried to just look at Golgi, peroxisome, endosome (and perhaps lipid droplet, which seems to be missing entirely from any discovery here by the method), how does PIFiA compare to DeepLoc or Paired Cell InPainting? (In theory I can imagine the SPB will look different than endosomes or COP vesicles upon rotations of 90 degrees for example, but only if a convolutional filter large enough to associate pixels from opposite sides of the cell are employed)

4. The Methods section implies that imaging parameters were held constant from one GFP-labeled strain to the next (in terms of parameters such as the exposure times, laser intensities, etc.). As each gene in the GFP library is under control of its native promoter, however, there is substantial gene-to-gene variability in expression level. Assuming each strain was imaged with the same settings one would expect substantial strain-to-strain variability in pixel intensities; in the worst case scenario this could lead to pixel patterns that are unrelated to true protein localization. Does cluster assignment depend on pixel intensity in any way (e.g., what fraction of proteins are misassigned to cytoplasm because they are too dim to produce a pattern that would indicate their localization to a specific organelle?).

Minor comments/questions:

1. I believe the original Huh et al collection contained 4159 strains, here the collection assayed is down to 4049. Are there any systematic biases in what was excluded?

2. The tdTomato (nucleus) and E2-Crimson (cyto) markers would be predicted to have some spectral overlap when excited by 561nm light and filtered through a 600/40 filter cube (i.e., cytoplasmic signal would contaminate what is thought to be purely nuclear). To what degree does this result in misinterpretation of nuclear vs cytoplasmic localization interpretations (eg, on dual localization, etc)?

3. How reliable is the data from "Ybr016w, which is a tail-anchored plasma membrane protein that is orthologous to human CYSTM1", wouldn't the tail anchor be obscured by the C-terminal GFP tag?

4. It is unclear why PIFIA outperforms the very similar method of Paired Cell InPainting, is there any intuition that can be gleaned from this to help guide future design of networks for use in unsupervised classification?

5. In the text supporting Fig. 3f it is stated: "The ER-5 protein, Kre1, shows an ER localization but it also concentrates more discretely at the cell periphery compared to Ubx2" What is the basis for this conclusion? Analyzing the imaging data by eye does not show an obvious difference (unlike the case of Ftr1 and Tcb2, which is very clear).

Very minor comments (these have no bearing on the overall evaluation of the manuscript, hopefully the authors find them useful):

1. "label-free resolution of cellular spatial organization" - the authors might want to reword this to avoid confusion about whether this is label-free imaging or not (it is in the machine learning sense, but it isn't in the optical microscopy sense)

2. "We manually labeled 800 crops of cells from 103 different proteins, corresponding to distinct cell cycle stages (with 200 crops from each class)". Perhaps I missed it, but what was the criteria used for assigning to a given stage in the cell cycle? Was it simply based on bud emergence?

**Donnelly Centre**

for Cellular + Biomolecular Research

UNIVERSITY OF
TORONTO**Brenda J. Andrews, Ph.D., C.C., FRSC**

University Professor

Canada Research Chair in Systems Genetics & Cell Biology

The Donnelly Centre

Department of Molecular Genetics

University of Toronto

December 21, 2023

Maria Polychronidou, PhD

Senior Editor, Molecular Systems Biology

m.polychronidou@molsystbiol.org

Dear Maria,

Thank you very much for overseeing the recent review of our paper entitled "PIFiA: Self-supervised Approach for Protein Functional Annotation from Single-Cell Imaging Data". We appreciate the thoughtful comments of the two reviewers, and have addressed their concerns and suggestions in full in our revised paper.

Below, we detail how we responded to each of the reviewers' comments which we include in blue text, followed by our response in black text. We **hope you agree our manuscript is now ready for publication, and look forward to hearing from you.**

Yours sincerely,

Brenda J. Andrews, Ph.D., C.C., FRSC

University Professor

Canada Research Chair in Systems Genetics & Cell Biology

The Donnelly Centre

University of Toronto

160 College Street, Suite 1308

Toronto, Ontario M5S 3E1

brenda.andrews@utoronto.ca | 416-978-8562<http://www.thedonnellycentre.utoronto.ca/>**Detailed response to reviewers****Reviewer #2:**

The reviewer notes the high quality of our efforts to evaluate the performance of the PIFiA CNN, and that the characterization of protein localization heterogeneity using PIFiA's single cell representations is 'new and creative'. We appreciate these positive comments about our work.

The reviewer expresses a general concern regarding PIFiA's 'limited methodological novelty', and recommends we focus on the resource and downstream analyses as the main contribution as opposed to the method. We agree with this general comment and have attempted to convey the importance of the downstream pipeline in our paper. We have revised the paper to ensure

that the PIFiA neural network architecture is more appropriately considered as suggested by the reviewer and described in the detailed comments below.

Major comments

1. *The method is highly similar to the one presented in <https://doi.org/10.1101/2022.08.12.503783> (which the authors should cite) that uses the same concept. Cytoself (<https://doi.org/10.1038/s41592-022-01541-z>) that was applied to a similar task of encoding protein localization profiles. The authors' main justification for their design choices was the argument that optimizing the pretext task, without enforcing reconstruction, creates better representations. This argument is made in page 3-4 ("A disadvantage of autoencoders is that they often learn features that are not relevant to protein morphology and localization, such as cell position, imaging artifacts and noise. Also, pixel-level reconstruction is computationally expensive and often unnecessary for representation learning. These issues reflect the autoencoder's training objective, which targets identical reconstruction of the input."). This argument was evaluated directly in the Cytoself paper, where it was shown that the reconstruction (decoder) contributes to effective representations (see Fig. 4 "Cytoself without decoder" copied below). The comparisons of PIFiA to other approaches (Cell Profiler, DeepLoc, Paired Cell Impainting, DeepLoc + PIFiA) are not sufficient to make this point, which should be evaluated explicitly by comparing the same model with and without reconstruction. If the authors want to make this argument, I recommend evaluating Cytoself with vs. without decoding (I am aware of other labs who were successful at using Cytoself's representations to analyze their data), or at least PIFiA, with vs. without decoding. If the decoder indeed deteriorates the representations then these conclusions, contradicting those of Cytoself, should be discussed.*

These are important points, and we have addressed them in our revised paper:

a) We performed a comparison between PIFiA and cytoself, which is now included in **Figure 2**. We used the cytoself version from the Kobayashi et. al. 2022 GitHub repository (more details in the Methods section). We found that PIFiA achieved superior performance on four benchmarks of functional module detection compared to cytoself, with cytoself performing on par or slightly above the second-best performing self-supervised approach, Paired Cell Impainting.

We say (p 6, 3rd paragraph):

"Also, PIFiA outperformed *cytoself*, a self-supervised approach that utilizes a VQ-VAE in its architecture, which achieved similar performance to Paired Cell Impainting."

Given the focus of our introduction, the reviewer suggested that an evaluation of Cytoself with vs. without decoding for comparison to the PIFiA architecture. As noted by the reviewer, we agree that removing the decoder from autoencoder-based architectures would result in lower performance, mainly for two reasons: 1) its representations were not directly trained on a pretext task unlike final-layer representations, and hence are less suitable for downstream tasks and; 2) its representations would have lower dimensionality, resulting in less precision and worse performance. Our main goal was to simplify the architecture and training pipeline without sacrificing quality. Hence, we have de-emphasized and re-arranged the part of the introduction that discusses autoencoders.

Our method was developed with the aim of minimizing architecture complexity and improving efficiency, while achieving strong performance on downstream tasks. We proposed to use a classic and simple convolutional network, previously used in DeepLoc work (Kraus et al., 2017), as opposed to more complex autoencoder-based models, and achieved state-of-the-art performance on a range of downstream tasks. We also want to note that we put more focus on the downstream analysis components of the PIFiA pipeline (which spans a wide range of biological standards (protein complexes, bioprocesses, localizations and pathways), and less emphasis on neural network architecture and hyperparameter optimization, in contrast to Kobayashi et. al. work. Overall, as noted by both reviewers, the main contributions of our paper are comprehensive downstream analysis and novel approaches for functional module discovery from deep learning-based representations of microscopy data.

2. *Generalization across biological replicates. Is it necessary to learn new representations for each dataset or could learned representations effectively generalize across different datasets? This is an important point that is crucial for understanding the method's capacity and potential use by others, but is not evaluated in the manuscript. The dataset contains two biological replicates for each protein (with four fields of view per replicate). The authors could train PIFiA on one biological replicate and evaluate it on the other (and vice versa) to evaluate this point. Whether these representations can be transferred to other datasets should be discussed. It was demonstrated that Cytoself can perform well on an external independent dataset - this can be referenced in the discussion in this context*

We thank the reviewer for their suggestion. Following this advice, we performed a generalization analysis for PIFiA using two unseen datasets of yeast cell images: **YeastRGB** (B. Dubreuil et. al., 2019; <https://www.ncbi.nlm.nih.gov/pmc/articles/PMC6324022/>) and **CYCLoPS** (Collection of Yeast Cells Localization Patterns, J. Koh et. al., 2015; <https://pubmed.ncbi.nlm.nih.gov/26048563/>).

Both datasets contain images of >4,000 unique GFP-tagged proteins. For both collections, we used 64x64 px single-cell crops using only the GFP channel, standardized each image as in PIFiA pipeline (see Methods) and passed the crops through the network to obtain feature profiles. We averaged all single-cell feature profiles of each protein in the dataset, obtaining aFPs, and visualized the averaged per-protein feature profiles. We show our results with 2D t-SNE colored by localization categories from Huh et. al. (**Fig. EV3**).

Our results are summarized in **Figure EV3** and we also added a new subsection to the results - **Generalization to unseen datasets**, p 8:

Generalization to unseen datasets

We investigated the generalization capabilities of the PIFiA neural network by applying it to previously unseen yeast imaging datasets. Specifically, we evaluated the performance of PIFiA on the publicly available CYCLoPS⁶⁹ and YeastRGB⁷⁰ datasets. Both datasets contain images of >4,000 unique GFP-tagged proteins: CYCLoPs images show a version of the ORF-GFP collection and YeastRGB contains images of a new collection based on a different fluorescent protein, mNeonGreen. We applied the PIFiA network out-of-the-box to single-cell crops of the fluorescently-tagged protein images from these datasets and extracted their feature profiles (see Methods for details on feature extraction).

We generated aFPs from both datasets and used tSNE to visually represent the similarity between feature profiles, with points in close proximity reflecting similar visual characteristics. (**Fig EV3**). We color-coded the tSNE maps according to the annotated subcellular localizations of the yeast cell components using a well-known standard of yeast protein localization¹¹. This visualization strategy allowed us to observe the formation of dense clusters corresponding to specific subcellular compartments. Nucleus, cytoplasm, mitochondrion, ER, vacuole and nucleolus were the most distinguishable localizations across both datasets (**Fig EV3A, B**). Thus, the PIFiA network showed strong generalization capabilities on two unseen datasets without any fine-tuning of its weights. We attribute this generalization to diverse data augmentation that was applied to the training data. Overall, our results confirm the feasibility of applying PIFiA for the analysis of novel datasets.

3. *The authors do not explicitly state whether the ORF-GFP yeast imaging dataset itself is one of the key contributions of this manuscript. Was the data collected for this manuscript? Would the raw and the processed data become publicly available? The visualization is a nice contribution, but the link to the files https://thecellvision.org/pifia_files is not working.*

Yes, we created this dataset as part of this manuscript, and we describe how we made the collection on page 5 in the results and in the methods. We have fixed the link and the raw and processed data are now publicly available through the CellVision website (https://thecellvision.org/pifia_files/).

4. *The text in the figures is tiny, which makes some of them hard to follow, especially Figures 3 and 5.*

We increased the size of subfigures, subtitles and labels in all our figures. We specifically focused on Figures 3 and 5 to make the text easily readable, and increased brightness and saturations of the t-SNE color schemes.

5. *I was not able to fully follow the results reported in Figure 5. More details in the main text would be helpful.*

We agree that Figure 5 is complex and we added clarifications to the subsection **Prediction of functional modules using PIFiA single-cell feature profiles** in the Results section, p 12:

We showed that protein-level feature profiles, or aFPs, can present a range of microscopy patterns in a compressed numerical form, which can be used for clustering and building hierarchical dendrograms. Using AMI scores at different correlation thresholds we were able to resolve functional information associated with hierarchical clustering of PIFiA aFPs (**Fig 2A**) and determine an optimal correlation threshold for discovering functional modules, such as protein complexes (**Appendix Fig S1A**). However, averaging feature profiles leads to information loss, which is not optimal for more precise analysis. Hence, we explored the use of single-cell feature profiles for the identification of functional modules. In particular, we focused on whether we could use scFPs for improved identification of protein complexes, which represent functional modules whose components are expected to colocalize within a single cell.

We derived our scFPs clustering analysis from a straightforward intuition - scFPs belonging to the same protein or the same protein complex should be indistinguishable, given the resolution limits of light microscopy. To visually illustrate this hypothesis on protein complex member distributions with scFPs, we projected scFPs from the test set using 2D tSNE (**Fig 5A**).

We also updated the Figure 5 legend with more details.

Other comments and suggestions:

1. *A more detailed explanation of the comparison to DeepLoc + PIFiA would be helpful. In page 19, the authors state that the benchmark was "re-trained on a larger set of images with less accurate protein-level labels instead of expensive yet more precise single-cell labels." - what is the meaning and implications of "less accurate labels"? Is it due to the human protein localization annotations?*

Thank you for bringing up this point. The original DeepLoc neural network was trained on a small dataset (21,882 images) of human-labelled single cell crops (See Kraus et al., 2017). In this study, we used a different dataset and, hence, we expanded the human localization labels from Huh et al. to all single cell crops of the same protein within this new dataset. While such a labeling method is much more productive (1,432,774 images) compared to manual labeling of each single cell crop, it inevitably causes some level of mislabeling, especially in case of the multi-compartment localization. That is why we refer to these labels as "less accurate labels". We compared PIFiA to both the original DeepLoc (out-of-the-box) and our version that was re-trained on our new dataset.

We added this clarification to the mentioned subsection of Methods, p20.

We used the DeepLoc model by Kraus et. al.⁸ as our supervised learning baseline. We investigated two training modes: a) with per-protein localization labels from Huh et. al.; b) with single-cell localization labels manually annotated in our lab. Per-protein annotations can sometimes be subjective and might not capture the nuances of protein localization at a single-cell level accurately. Such annotations apply to all of single-cell images of a protein and are available for a significant part of the yeast proteome from the Huh et. al. study. In contrast, single-cell labels were manually annotated for the individual cell images by research scientists in our lab. While more precise, such labels are expensive and encompass a much smaller portion of the proteome. The original DeepLoc version uses single-cell labels. For a fair comparison, we used both single-cell and protein-level localization labels to train DeepLoc and reported the corresponding results.

2. *Moreover, to ensure a fair comparison between the weakly/self supervised approach and supervised approaches, it is advisable to consider using the same PIFiA architecture for supervised training. While comparing DeepLoc+PIFiA as a benchmark is a valuable evaluation, it remains unclear whether the observed difference in results is derived from the difference in architectures or training. Thus, I suggest:*
- A more clear and comprehensive description on the training labels for the supervised approach. What were the limitations/noise in these labels?*
 - Compare PIFiA trained with weak labels versus trained with protein-level labels, using the same architecture.*
 - Assuming that the focus of the manuscript will not be the Method or that the authors will perform the Cytoself evaluation with/without the decoder, these evaluations are not necessary.*

As suggested by the reviewer (and addressed above), we have shifted the focus of the manuscript towards biological discoveries and evaluations, not the method itself. However, we have addressed the above points in our revised paper:

- We added a more comprehensive description of the training labels for supervised approaches to the manuscript (Methods, page 20, see point 1, above). Overall, we used two sets of supervised labels: single-cell localization labels (annotated in our lab per cell, more precise but low quantity), and per-protein labels from Huh. et. al. study (annotations are per protein, high quantity but low precision).
- We want to clarify that in all our experiments, we are using the same backbone architecture for PIFiA and DeepLoc. Hence, the analysis in Fig. 2 is a valid comparison of the training objectives under the same neural network architecture. Additionally, we found that PIFiA's performance does not change substantially when adding more layers to the architecture (Fig. EV1).
- We have reduced the emphasis on the method part of our manuscript and focused more on the results and downstream analysis. We rearranged parts of the Introduction and Results, as described above.

3. *Fig. 2 legends, please provide explicit references to Kraus et al., and Huh et al.*

Thank you, we have added the references.

4. *Fig. 2. It would be helpful to plot the random prediction accuracy (horizontal line) as a baseline for context.*

We added performance lines for a randomly initialized network (with PIFiA architecture) to Figure 2.

5. Fig. 3a - I recommend excluding the features visualization which does not provide any useful information.

We have found that some biologists like the clustergram visualization, as it is more intuitive and helps the reader understand the rest of the figure. As such, we have left part a) in the figure.

6. Fig. 3d - the text and legend mentions KDE visualization, but contours are not present in the figure. In general, I would recommend revising Fig. 3 legend.

To improve the clarity of visualization, considering that we are showing 28 bioprocesses that are close to each other on tSNE, we opted for blurred-contour KDE visualization instead of contour lines. We amended Figure 3 legend to clarify this.

7. I found the images of the gradients in Fig. 6d not very convincing. What can be learned from regions with higher intensities that tend to have higher gradients, how does this support the claim made that "the CNN captures a variety of within-cell morphological patterns"

We want to note that a gradient map is one of the ways to understand individual features and their relationship to the input image, which is often not very accurate due to individual image variance. While regions with higher intensity are expected to have higher gradient values (unless they are not meaningful for the training objective), the interesting observation here is that different parts of the high-intensity regions are meaningful for each particular feature.

We added this clarification to the Results section, p 15, 2nd paragraph.

We observed that different features of the same image resulted in different gradient maps. While regions with higher intensity often have higher gradient values (unless they are not meaningful for the training objective), the interesting observation here is which part of these regions are meaningful for the particular feature. For example, the last gradient map of Pex3-GFP protein shows punctate localization patterns with three distinct punctae. Interestingly, different features react to different parts of the image - feature 4 reacts to the lower dot on the image, and feature 23 reacts to two other dots.

8. Comparison to Cytoself

a. Cytoself was demonstrated for downstream analysis and interpretation. This should be cited in this context (page #4 in the introduction). One example is Fig. 3b which is very similar to the spectral representations (Fig. 5) in Cytoself.

b. In the Discussion that authors say that "A similar approach, cytoself, was used to visually separate protein complexes from different compartments in human cells on tSNE.", downplaying the contribution of the Cytoself paper (which did more than just tsne visualization). Please report these contributions properly.

- a) We have cited the cytoself paper in the intro and figure legends, and added a comparison of PIFiA performance to cytoself in Figure 2 as suggested by the reviewer.
- b) We have edited the discussion to more accurately reflect the contribution of the cytoself paper, p16, last paragraph.

A similar approach, *cytoself*²¹, was used to visually separate protein complexes from different compartments in human cells. Like the PIFiA pipeline, its self-supervised training scheme requires no preexisting knowledge or categories, allowing it to reveal a highly resolved protein subcellular localization atlas that summarizes the major scales of cell

organization²¹. Both the cytoself study and our work validate image-based feature profiles for downstream studies of protein organization and function in eukaryotic cells, both yeast (this work) and human cells²¹. Notably, we demonstrate that PIFiA can distinguish protein complexes from the same compartment in yeast cells, which are 5 to 30-fold smaller in size than human cells, providing a quantitative approach for downstream analysis and identification of functionally related proteins.

9. In the Discussion that authors refer to "in silico colocalization" a term that was never defined / explained.

We added a definition and explanation of this term to the Discussion section, p17.

In essence, the approach performs *in silico* colocalization, when two or more biological entities, such as proteins, are analysed for similarity based on their respective localization patterns or positions within a cell⁷¹. In contrast to experimental colocalization, which is time-consuming and expensive, *in silico* colocalization can be performed within seconds for multiple proteins at a time. In the case of PIFiA, this can be achieved not only with high speed but also with remarkable precision.

10. Fig. S6 (cell cycle) is not referenced in the text.

Corrected, this is now Appendix Fig S2 – thank you.

Reviewer #3

Major questions

1. One of the most surprising findings to me was the ability of PIFiA to distinguish diffraction-limited structures from each other given that structure was detected through CNN's. Do the authors have an explanation for why this method is effective in distinguishing peroxisomes or endosomes from COP vesicles for example (e.g., in Fig 5)? The authors went to some lengths to try and open the "black box" in Fig. 6 (an important exploration for which the authors are to be commended), but I think PIFiA would be even better tested by examining how subdiffraction structures are distinguished from each other. As a concrete suggestion (though the authors will undoubtedly come up with a better idea): what kind of feature profiles are learned upon masking of pixels contained within the brightest puncta when looking at peroxisomal (or Golgi or lipid droplet, etc) proteins? Do these feature profiles exhibit any plausible biological relevance (e.g., if one masks out the punctate signals from peroxisomal proteins, would gradient maps as performed in Fig. 6d have overlap with the ER, suggesting that PIFiA is able to pick up on subtle protein trafficking patterns to help it distinguish apparently diffraction limited mature structures?). In the case of peroxisomes, though, it appears not to be the case (e.g., in Fig. 4f there does not appear to be shared signal between peroxisomes and ER).

We agree with the reviewer that the ability of PIFiA to produce feature profiles that contain sufficient information to distinguish 'diffraction-limited' structures is a useful and powerful feature of the method.

Our first attempt at automated identification of protein localization in yeast images used an ensemble of binary classifiers (Chong 2015), and we had difficulty distinguishing small punctate compartments. However, our deep CNN, DeepLoc (Kraus 2017), was able to classify proteins in these compartments with 0.7-0.85 precision, so we were not surprised that PIFiA could distinguish them. According to importance weights of individual features, it is likely that distinction between punctate structures is driven by slightly different punctate patterns characterizing each compartment (e.g. size of puncta, dots distribution etc.). For instance, from Fig. 6A we can observe that both features 50 and 15 are important for punctate structures identification. Feature 50, however, is more pronounced in Golgi than in peroxisome, while feature 15 is very pronounced in peroxisome.

We have added an example showing how peroxisomes are recognized, p 15, 2nd paragraph.

We observed that different features of the same image resulted in different gradient maps. While regions with higher intensity often have higher gradient values (unless they are not meaningful for the training objective), the interesting observation here is which part of these regions are meaningful for the particular feature. For example, the last gradient map of Pex3-GFP protein shows punctate localization patterns with three distinct punctae. Interestingly, different features react to different parts of the image - feature 4 reacts to the lower dot on the image, and feature 23 reacts to two other dots.

Removing bright pixels from the image to inspect gradient maps can result in altered prediction, because the neural network takes into account the combination of all input patterns rather than looking at each individual pattern separately. Hence, by removing a key structure, a different set of pixels may be activated because the image now presents a new pattern to the network.

2. The other major potential of PIFiA as a discovery tool is its potential ability to use single cell feature profiles to assign proteins to functional modules based on subtle yet highly correlated feature profiles. These subtle features could represent substantial new insight into functionally relevant protein-protein interactions. But for this to be the case the authors should experimentally test how these subtle features change upon perturbations to the putative protein-protein interactions. The ideal type of perturbation would not be one that induces a gross mislocalization of a protein, but instead one that only weakens the physical association between two proteins in a complex and results in disruption to the subtle aspects of the feature profile correlations. As an example of my suggestion, Eugster et al EMBO J 19: 3905-3917 2000 (doi: 10.1093/emboj/19.15.3905) contains a dissection of COP I domains whose genetic ablation induces changes to COP architecture; the authors might know better examples where mutations disrupt binding surfaces between proteins that cluster together in the same functional module in PIFiA and could be employed to test their effect on subtle feature correlations. Alternatively, the authors could provide an analysis of what fraction of proteins known to belong to functional modules but whose memberships in these modules are not successfully uncovered by PIFiA feature correlations to provide important context into the limitations of the method.

The reviewer proposes some interesting experiments. For this paper, we performed the analysis suggested by the reviewer exploring what functional protein modules we can detect with PIFiA. We have added p 14, 2nd paragraph:

To provide some context into the limitations of the method, we looked at what types of protein complexes were under-represented in our set of high-confidence predictions. Identification by PIFiA was independent of protein abundance or number of members in the complex. The protein complexes we identified were enriched for localizations in small organelles and compartments (peroxisome, actin, nuclear periphery, endosome, spindle pole) and depleted for those in large diffuse compartments such as nucleus and cytoplasm. We note that not all protein complexes will have a distinct localization. Using PIFiA features we have identified protein complexes from most compartments in the cell.

3. I imagine much of the measured success of PIFiA (in terms of aggregative metrics such as the F-score, AP, etc) lies in many successfully classified proteins belonging to morphologically distinct organelles (eg ER, mito, cytoplasm). To more finely gauge how much of an advance PIFiA is compared to the other methods tested here I would recommend reproducing Fig. 2 for cases where morphologies are not dramatically different. For example if the authors tried to just look at Golgi, peroxisome, endosome (and perhaps lipid droplet, which seems to be missing entirely from any discovery here by the method), how does PIFiA compare to DeepLoc or Paired Cell InPainting? (In theory I can imagine the SPB will look different than endosomes or COP vesicles upon rotations of 90 degrees for example, but only if a convolutional filter large enough to associate pixels from opposite sides of the cell are employed)

Thank you for bringing up this point. We want to note that F-score and AP reflect a combination of precision and recall, meaning that high score requires 1) proteins with similar feature profiles to belong to the same localization (precision), and 2) proteins from the same localization to belong to the same cluster (recall). This is different from classification, where only a correct

localization assignment affects the score, but feature similarity is not taken into account. While these scores to some extent reflect performance on similar morphologies (because the validation set contains proteins from similar-looking organelles and their misannotation by the network would lead to a lower score), we agree that comparing performance specifically on the similar compartments is an interesting analysis. Hence, we performed the analysis on solely Golgi, endosome and peroxisome-localized proteins. We included aFPs for proteins with a single Huh et al. annotation and performed an analysis similar to Figure 2, and the performance (Average Precision) is summarized below:

	GO CC	GO BP
PIFiA	0.13	0.06
Paired Cell Inpainting	0.09	0.03
DeepLoc original	0.08	0.03
DeepLoc + PIFiA	0.1	0.05

We can see that while overall performance is slightly worse than when whole-proteome metrics are computed, which is expected since this task is harder, PIFiA outperforms other approaches. We added this information to the methods comparison, p6:

To evaluate PIFiA performance on compartments with similar morphologies, we did an extra evaluation run similar to Fig 2A, but only including aFPs of proteins from Golgi, endosome and peroxisome (with a single Huh et. al. localization label). We obtained 0.13, 0.09, 0.08 and 0.1 AP scores for PIFiA, Paired Cell Inpainting, DeepLoc original and DeepLoc+PIFiA, confirming that PIFiA is capable of distinguishing proteins from similar-looking morphologies.

We also investigate more subtle standards of subcellular localization, such as pathways, bioprocess and protein complexes in Figure 2.

4. The Methods section implies that imaging parameters were held constant from one GFP-labeled strain to the next (in terms of parameters such as the exposure times, laser intensities, etc.). As each gene in the GFP library is under control of its native promoter, however, there is substantial gene-to-gene variability in expression level. Assuming each strain was imaged with the same settings one would expect substantial strain-to-strain variability in pixel intensities; in the worst case scenario this could lead to pixel patterns that are unrelated to true protein localization. Does cluster assignment depend on pixel intensity in any way (e.g., what fraction of proteins are misassigned to cytoplasm because they are too dim to produce a pattern that would indicate their localization to a specific organelle?).

Since GFP intensities are certainly part of the feature profiles, cluster assignment may depend on pixel intensity. We explore this in Table EV2 by showing the median and SD for GFP intensity for each cluster (Summary of each group). We have added to the text (p 9, bottom paragraph):

Third, as expected, some of the groupings appeared to be based on protein features that were easily discernible. Proteins in some sub-compartmental groups have a tight distribution of GFP intensities, suggesting that abundance is likely an important feature for that group. For example, *nuc-1* clustering likely resulted from high protein abundance, and included histones and metabolic enzymes (median GFP intensity *nuc-1* proteins=5834±2103 vs median for all *nuc* proteins=745±793) (**Table EV2**). Likewise,

nuc-3 proteins all had low abundance (median GFP intensity=678±52) and this group was enriched for mitotic nuclear division and chromosome segregation. For some of the other groups, clustering appeared to result from differences in the spatial distribution of pixels in a region.

Minor comments/questions:

1. I believe the original Huh et al collection contained 4159 strains, here the collection assayed is down to 4049. Are there any systematic biases in what was excluded?

We have added to the Methods, p 18. Construction of mutant arrays for imaging:

We successfully constructed strains with 4049 (97.4%) GFP-tagged genes out of 4156 strains in the ORF-GFP collection. The missing GFP strains include those in linkage groups for *CAN1* and *LYP1*, as occurs in all SGA-derived collections. Other missing strains were those we were unable to grow from our original stocks. The missing strains have no bias for protein abundance (from Ho et al., 2018 PMID: 29361465). BY GO cellular component, they are enriched for mitochondrial *cytochrome complex* (6 genes; Bonferroni-corrected P value=0.000368).

2. The *tdTomato* (nucleus) and *E2-Crimson* (cyto) markers would be predicted to have some spectral overlap when excited by 561nm light and filtered through a 600/40 filter cube (i.e., cytoplasmic signal would contaminate what is thought to be purely nuclear). To what degree does this result in misinterpretation of nuclear vs cytoplasmic localization interpretations (eg, on dual localization, etc)?

PIFiA was trained on the GFP channel, so the RFP markers should not influence the localization calls. The red channel, with nucleus and cytosol marked, was only used in the cell cycle analysis.

3. How reliable is the data from "*Ybr016w*, which is a tail-anchored plasma membrane protein that is orthologous to human *CYSTM1*", wouldn't the tail anchor be obscured by the C-terminal GFP tag?

We thank the reviewer for pointing this out. Yes, *Ybr016w* is actually a *predicted* tail-anchored membrane protein (PMID: 12514182); however, unlike most of the other predicted tail-anchored proteins, it has the same localization with N- and C-terminal GFP, suggesting that the prediction may not be correct.

We had just wanted to make the point that our analysis puts several uncharacterized proteins in *CP-1*, but we do not want to distract the reader about the mechanism of localization, so we have removed all comments about putative roles and changed the sentence to read (p 10, 2nd last paragraph):

"In addition to the MDR transporters, the *CP-1* group contains 14 novel mother-specific proteins, including several other transporter proteins (*Atr1*, *Ftr1*, *Hxt6*, *Mep1*, *Mep3*, *Qdr2*, and *Qdr3*), proteins with roles in signaling (*Gpa2*, *Mid2*, *Psr1*), and three uncharacterized proteins (*Ybr016w*, *Ina1*, and *Crp1*; **Table EV2**)."

4. It is unclear why PIFiA outperforms the very similar method of Paired Cell InPainting, is there any intuition that can be gleaned from this to help guide future design of networks for use in unsupervised classification?

Paired Cell Inpainting uses a training objective of reconstructing a GFP channel on the query image (with masked GFP channel), while PIFiA is trained to classify proteins based on an input image. Reconstruction is often prone to noise and variance between input images, while protein classification is a more robust objective that forces the network to learn relevant features for downstream analysis. We added this clarification to the introduction.

5. In the text supporting Fig. 3f it is stated: "The ER-5 protein, Kre1, shows an ER localization but it also concentrates more discretely at the cell periphery compared to Ubx2" What is the basis for this conclusion? Analyzing the imaging data by eye does not show an obvious difference (unlike the case of Ftr1 and Tcb2, which is very clear).

Some of our sub-compartmental groups had obvious differences from others with the same localization but most were more subtle and we are trying to be clear about that. We have added to the text, p10:

"We show three examples of pairs of proteins from different sub-compartmental groups that vary in their localizations to different extents."

We go on to say:

"The difference between these is more subtle: the ER-5 protein, Kre1, shows an ER localization but also an increased concentration at the cell periphery compared to Ubx2, as can be seen in the overlay (**Fig. 3F**, bottom row). This localization difference was observed in other members of these sub-compartmental groups, with ER-3 proteins tending to have a more diffuse localization and ER-5 proteins localizing more at the cell periphery (**Fig EV5**).

Very minor comments (these have no bearing on the overall evaluation of the manuscript, hopefully the authors find them useful):

1. "label-free resolution of cellular spatial organization" - the authors might want to reword this to avoid confusion about whether this is label-free imaging or not (it is in the machine learning sense, but it isn't in the optical microscopy sense)

We have reworded the sentence. p 7, para3

"Our clustering analysis showed that PIFiA aFPs capture information from cell images that enables unsupervised resolution of cellular spatial organization, grouping proteins by shared localization and biological function (**Fig 3A,B**)."

2. "We manually labeled 800 crops of cells from 103 different proteins, corresponding to distinct cell cycle stages (with 200 crops from each class)". Perhaps I missed it, but what was the criteria used for assigning to a given stage in the cell cycle? Was it simply based on bud emergence?

This is correct - we manually labeled crops according to bud emergence from cytosolic & nuclear channels. We clarified this part in Methods.

14th Feb 2024

Manuscript Number: MSB-2023-11946R

Title: PIFiA: Self-supervised Approach for Protein Functional Annotation from Single-Cell Imaging Data

Dear Brenda,

Thank you for sending us your revised manuscript. We have now heard back from the two reviewers who were asked to evaluate your revised study. As you will see below, the reviewers are satisfied with the modifications made and they support publication. Reviewer #2 recommends some text modifications which we would invite you to perform in a final round of minor revision. We would also ask you to address some remaining editorial issues listed below.

- Our data editors have noted the following points that need to be fixed in the figure legends:
 - Figure panel 5e is not labelled in the figure, however the legend for the same is labelled as 5e. This needs to be corrected.
 - Information related to n is missing in the legends of figures 2a-b, e; EV 1c.
 - The error bars need to be defined in the legends of figures 2a-c, e; EV 1c.
- The funding information provided in the manuscript text (Acknowledgements) should match the information entered in the online submission system. Currently, the "Vector Institute Postgraduate Affiliate Scholarship" is missing from the submission system.
- The References should be formatted according to the Molecular Systems Biology reference style (i.e., ordered alphabetically and listing the first 10 authors followed by et al).
- Please include a "Disclosure and Competing Interests Statement" in the main text.
- Please remove the 'Authors Contributions' from the manuscript. The 'Author Contributions' section is replaced by the CRediT contributor roles taxonomy to specify the contributions of each author in the journal submission system. Please use the free text box in the 'author information' section of the online submission system to provide more detailed descriptions if needed (e.g., 'X provided intracellular Ca⁺⁺ measurements in fig Y').
- We would ask you to make the image data available at the Image Data Resource (<https://idr.openmicroscopy.org>) or at the BioImage Archive (<https://www.ebi.ac.uk/bioimage-archive>).
- Please include callouts to Fig. 2D-E in the main text.
- Tables EV1-EV4 should be renamed to Dataset EV1-EV4 and their callouts in the text should be updated.
- The synopsis image does not display well at the final required size. Please resupply the image as a jpg or png at the required final size (it needs to be exactly 550 px wide, and the height ideally < 500 px), ensuring that all labels are legible. Reorganizing the image so that not all image elements are displayed on "one line" might work better.

Please resubmit your revised manuscript online, with a covering letter listing amendments and responses to each point raised by the referees. Please resubmit the paper ****within one month**** and ideally as soon as possible. If we do not receive the revised manuscript within this time period, the file might be closed and any subsequent resubmission would be treated as a new manuscript. Please use the Manuscript Number (above) in all correspondence.

Click on the link below to submit your revised paper.

Kind regards,

Maria

Maria Polychronidou, PhD

If you do choose to resubmit, please click on the link below to submit the revision online before 15th Mar 2024.

IMPORTANT:

Please note that corresponding authors are required to supply an ORCID ID for their name upon submission of a revised manuscript (EMBO Press signed a joint statement to encourage ORCID adoption). (<https://www.embopress.org/page/journal/17444292/authorguide#editorialprocess>)
Currently, our records indicate that the ORCID for your account is 0000-0001-6427-6493.

Link Not Available

***** PLEASE NOTE ***** As part of the EMBO Press transparent editorial process initiative (see our Editorial at <https://dx.doi.org/10.1038/msb.2010.72> , Molecular Systems Biology will publish online a Review Process File to accompany accepted manuscripts. When preparing your letter of response, please be aware that in the event of acceptance, your cover letter/point-by-point document will be included as part of this File, which will be available to the scientific community. More information about this initiative is available in our Instructions to Authors. If you have any questions about this initiative, please contact the editorial office (msb@embo.org).

Reviewer #2:

The authors have addressed my concerns. In my opinion, the manuscript is ready for publication and will be a nice contribution to the field.

There is one last issue of better placing this work in context of previous work that should be determined by the authors and editor. Moshkov et al (ref #14) demonstrated a very similar concept of training a CNN to predict compound labels and using the inner representation to encode the corresponding perturbation phenotypes. I recommend emphasizing this point in the Discussion rather than the less specific current statement.

Reviewer #3:

I thank the authors for their work in responding to the feedback offered by the reviewers. All minor points in my feedback were completely addressed. In my review, Major points 1 and 3 were comprehensively addressed as well and Major point 2 was adequately addressed as far as making clear where there are limitations to PIFiA. As far as Major point 4, I am still unclear about whether overall signal intensity will lead to the misclassification of proteins whose expression level is low (especially at risk are proteins being misclassified to the cytoplasm because their low signal intensity makes their spatial inhomogeneity hard to discern) but I thank the authors for compiling the data in a way where readers can judge this on their own: the assignments of proteins to PIFiA-assigned groups in Table EV2, plus the availability of the images (hopefully in TIFF or some other format that preserves raw signal intensities to enable user-defined analyses) will enable readers to check whether or not their protein of interest has been misclassified by PIFiA.

Therefore based on these reasons I do not believe there is any need to delay publication of this manuscript any further. I thank the authors again for sharing their work on this tool with us.

February 23, 2024
Maria Polychronidou, PhD
Senior Editor
Molecular Systems Biology

Dear Maria,

We are pleased that the reviewers of our paper entitled "PIFiA: Self-supervised Approach for Protein Functional Annotation from Single-Cell Imaging Data" were satisfied with our revisions and that they support publication. We are submitting the final revised version of our manuscript which addresses the remaining editorial issues and the text modification suggested by Reviewer #2.

Points noted by data editors:

-- Figure panel 5e is not labelled in the figure, however the legend for the same is labelled as 5e. This needs to be corrected.

We cannot find any evidence of Fig 5e in either the figure or the legend.

-- Information related to n is missing in the legends of figures 2a-b, e; EV 1c.

We have defined n in the legends of figures 2a-b, e and EV 1c.

-- The error bars need to be defined in the legends of figures 2a-c, e; EV 1c.

We have defined the error bars for Fig 2A-E and EV 1c in the legends.

- The funding information provided in the manuscript text (Acknowledgements) should match the information entered in the online submission system. Currently, the "Vector Institute Postgraduate Affiliate Scholarship" is missing from the submission system.

Corrected.

- The References should be formatted according to the Molecular Systems Biology reference style (i.e., ordered alphabetically and listing the first 10 authors followed by et al).

References are now formatted according to MSB style

- Please include a "Disclosure and Competing Interests Statement" in the main text.

Included

- Please remove the 'Authors Contributions' from the manuscript. The 'Author Contributions' section is replaced by the CRediT contributor roles taxonomy to specify the contributions of each author in the journal submission system. Please use the free text box in the 'author information' section of the online submission system to provide more detailed descriptions if needed (e.g., 'X provided intracellular Ca⁺⁺ measurements in fig Y').

Statement has been removed

- We would ask you to make the image data available at the Image Data Resource (<https://idr.openmicroscopy.org>) or at the BioImage Archive (<https://www.ebi.ac.uk/bioimage-archive>).

All image data are freely available at CellVision.org and we are in the process of also making

the data available at BioImage Archive (in our experience this can take a while due to the size of the dataset and people generally access our data from our dedicated website).

- Please include callouts to Fig. 2D-E in the main text.

We have added callouts to Fig. 2D-E in the main text.

- Tables EV1-EV4 should be renamed to Dataset EV1-EV4 and their callouts in the text should be updated.

All Tables have been renamed Datasets in main text and in the tables themselves.

- The synopsis image does not display well at the final required size. Please resupply the image as a jpg or png at the required final size (it needs to be exactly 550 px wide, and the height ideally < 500 px), ensuring that all labels are legible. Reorganizing the image so that not all image elements are displayed on "one line" might work better.

We have resupplied the synopsis image at 550 px wide.

Reviewer #2 comment

There is one last issue of better placing this work in context of previous work that should be determined by the authors and editor. Moshkov et al (ref #14) demonstrated a very similar concept of training a CNN to predict compound labels and using the inner representation to encode the corresponding perturbation phenotypes. I recommend emphasizing this point in the Discussion rather than the less specific current statement.

We have added two sentences to the Discussion describing the work of Moshkov

“Another study has implemented a similar representation learning paradigm to learn feature profiles of the images of cells with chemical or genetic perturbations¹⁴, where perturbations were used as training labels. Thus, perturbation phenotypes were learned as inner representations from the network, suggesting that such a training scheme is effective for various types of experiments and biological systems.”

Please let us know if any additional corrections/revisions are required, and that you for overseeing the review of our paper.

Yours sincerely,

Brenda J. Andrews, PhD., C.C., FRSC

University Professor

Canada Research Chair in Systems Genetics & Cell Biology

The Donnelly Centre

University of Toronto

160 College Street, Suite 1308

Toronto, Ontario M5S 3E1

brenda.andrews@utoronto.ca | 416-978-8562
<http://www.thedonnelycentre.utoronto.ca/>

28th Feb 2024

Manuscript number: MSB-2023-11946RR

Title: PIFiA: Self-supervised Approach for Protein Functional Annotation from Single-Cell Imaging Data

Dear Brenda,

Thank you again for sending us your revised manuscript. We are now satisfied with the modifications made and I am pleased to inform you that your paper has been accepted for publication.

Kind regards,

Maria

Maria Polychronidou, PhD
Senior Editor
Molecular Systems Biology
